# Glial Nrf2 signaling mediates the neuroprotection exerted by *Gastrodia elata* Blume in Lrrk2-G2019S Parkinson's disease

Yu-En Lin[1,2], Chin-Hsien Lin[3], En-Peng Ho[3], Yi-Ci Ke[3], Stavroula Petridi[4,5], Christopher JH Elliott[5], Lee-Yan Sheen[2]*, Cheng-Ting Chien[1,6]*

[1]Institute of Molecular Biology, Academia Sinica, Taipei, Taiwan; [2]Institute of Food Science and Technology, National Taiwan University, Taipei, Taiwan; [3]Department of Neurology, National Taiwan University Hospital, Taipei, Taiwan; [4]Department of Clinical Neurosciences and MRC Mitochondrial Biology Unit, University of Cambridge, Cambridge, United Kingdom; [5]Department of Biology and York Biomedical Research Institute, University of York, York, United Kingdom; [6]Neuroscience Program of Academia Sinica, Academia Sinica, Taipei, Taiwan

*For correspondence:
lysheen@ntu.edu.tw (L-YanS);
ctchien@gate.sinica.edu.tw (C-TC)

**Competing interest:** The authors declare that no competing interests exist.

**Abstract** The most frequent missense mutations in familial Parkinson's disease (PD) occur in the highly conserved *LRRK2/PARK8* gene with G2019S mutation. We previously established a fly model of PD carrying the *LRRK2-G2019S* mutation that exhibited the parkinsonism-like phenotypes. An herbal medicine, *Gastrodia elata* Blume (GE), has been reported to have neuroprotective effects in toxin-induced PD models. However, the underpinning molecular mechanisms of GE beneficiary to G2019S-induced PD remain unclear. Here, we show that these G2019S flies treated with water extracts of GE (WGE) and its bioactive compounds, gastrodin and 4-HBA, displayed locomotion improvement and dopaminergic neuron protection. WGE suppressed the accumulation and hyperactivation of G2019S proteins in dopaminergic neurons and activated the antioxidation and detoxification factor Nrf2 mostly in the astrocyte-like and ensheathing glia. Glial activation of Nrf2 antagonizes G2019S-induced Mad/Smad signaling. Moreover, we treated *LRRK2-G2019S* transgenic mice with WGE and found that the locomotion declines, the loss of dopaminergic neurons, and the number of hyperactive microglia were restored. WGE also suppressed the hyperactivation of G2019S proteins and regulated the Smad2/3 pathways in the mice brains. We conclude that WGE prevents locomotion defects and the neuronal loss induced by G2019S mutation via glial Nrf2/Mad signaling, unveiling a potential therapeutic avenue for PD.

## Editor's evaluation

Your study elucidates the molecular mechanisms by which a Chinese traditional medicine compound, GE, confers neuroprotection in mouse and fly models of LRRK2 G2019S-induced Parkinson's disease and provides an avenue for a potential therapeutic treatment for patients.

## Introduction

Parkinson's disease (PD) is a highly prevalent neurodegenerative disorder characterized by the loss of dopaminergic neurons in the substantia nigra projecting to the striatum of the basal ganglion, representing a circuit involved in motor planning and coordination. As a consequence, PD is associated

**eLife digest** Parkinson's disease is a brain disorder that leads to tremors and difficulties with balance and coordination. These symptoms are caused by the loss of neurons which release a chemical messenger that is needed to regulate movement called dopamine. Most treatments for this disease work by boosting levels of dopamine in the brain, but this can lead to severe side effects and these drugs often become less effective over time.

A traditional Chinese medicine called *Gastrodia elata* Blume (or GE for short) has previously been reported to relieve symptoms of Parkinson's disease in both human and animal studies when administered as a decoction or formula. However, it is unclear how GE protects dopamine-producing neurons and if this mechanism involves another type of brain cell known as glia that has also been linked to Parkinson's disease.

To investigate, Lin et al. studied fruit flies and mice that carry a genetic mutation that produces the symptoms and molecular characteristics of Parkinson's disease. The experiments showed that when the flies and mice were fed food containing water extracts of GE, they experienced less difficulties moving and had a higher number of intact dopamine-producing neurons.

Lin et al. found that GE switched on a protein in glial cells located near dopamine-producing neurons. Activation of this protein, called Nrf2, inhibited a signaling pathway in degenerating neurons that leads to the disease state. As a result, less dopamine-producing neurons were damaged and the animals' coordination and balance were maintained.

These findings suggest that GE could potentially provide an alternative or complementary therapy for Parkinson's disease, although it still needs to be studied further in humans. If the same effect is observed, the specific compounds in GE that have this protective effect could be isolated and analyzed to see if they could be used for treatment.

with motor abnormalities, bradykinesia, hypokinesia, rigidity, and resting tremor. Currently, the most frequently applied pharmacological treatment, levodopa (L-DOPA), exerts limited motor improvement and elicits negative side effects (*Ray Chaudhuri et al., 2018*). Hence, identifying and developing alternative or complementary treatments may assist in mitigating PD progression.

PD is a multicausal disease with a complicated etiology, including familial inheritance. More than 20 *PARK* genes have been genetically linked to PD, a number that is increasing (*Houlden and Singleton, 2012*). Missense mutations in *PARK8*, or *Leucine-rich repeat kinase 2* (*LRRK2*), induce characteristic PD symptoms and pathologies such as loss of dopaminergic neurons and the appearance of Lewy bodies (*Martin et al., 2014*). Notably, the most commonly observed mutation, dominant G2019S, among familial PD cases is located in the kinase domain of Lrrk2, which augments its kinase activity via auto- and hyperactivation (*Sheng et al., 2012*). The hyperactive G2019S mutant protein alters several cellular processes, including vesicle trafficking, microtubule dynamics, autophagy, mitochondrial function (*Martin et al., 2014*), and, most commonly, increases susceptibility to oxidative stress that contributes to neuronal degeneration (*Angeles et al., 2011*; *Nguyen et al., 2011*). These indicate that regulation of the hyperactivation of G2019S mutant protein appears to be a disease-modifying strategy.

Glia provide structural and metabolic supports to neurons and regulate synaptic transmissions, so they are important for the function and survival of dopaminergic neurons (*Lin et al., 1993*; *Sofroniew and Vinters, 2010*). Dysfunction in two major glial types, astrocytes and microglia, contribute to the onset and progression of both sporadic and familial PD (*Kam et al., 2020*). Astrocytes and microglia of postmortem PD brains exhibit pathological lesions and hyper-immunoactivity (*Miklossy et al., 2006*). A clinical trial involving downregulation of microglial oxidative stress highlights the significance of glia to PD (*Jucaite et al., 2015*). Moreover, Lrrk2 regulates the inflammatory response in microglia and the autophagy-lysosome pathway in astrocytes, with the G2019S mutation altering the size and pH of lysosomes (*Henry et al., 2015*; *Moehle et al., 2012*). Expression of G2019S mutant protein in neurons was previously shown to induce the secretion of Glass bottom boat (Gbb)/bone morphogenetic protein (BMP) signal that, in turn, upregulates Mothers against decapentaplegic (Mad)/Smad signaling in glia, which prompts feedback signals to promote neuronal degeneration (*Maksoud et al., 2019*). These studies indicate that G2019S mutant protein alters the homeostasis and interaction

between neurons and glia, contributing to PD pathogenesis. However, whether any dietary or pharmacological treatment blockading this neuron-glia interaction beneficiary to G2019S-induced PD is unclear.

Given that up to 70% of human *PARK* genes are conserved in the *Drosophila* genome, *Drosophila* is frequently used as a PD model for studying gene function, such as the *PARK1/SNCA* (*Chen and Feany, 2005*) and *PARK8/LRRK2* (*Liu et al., 2008*). Genetic and molecular linkage between *PINK1/PARK6* and *Parkin/PARK2* was first established in *Drosophila* (*Clark et al., 2006*). When overexpressed in *Drosophila* dopaminergic neurons, *LRRK2* transgenes carrying G2019S or other dominant mutations induce dopaminergic neuron loss and locomotion impairment, two age-dependent symptoms of PD (*Lin et al., 2010*; *Liu et al., 2008*). The G2019S model has further been used to screen a collection of FDA-approved drugs to suppress these PD phenotypes. Thus, *Drosophila* represents an amenable model of PD for genetic, molecular, and pharmacological study of potential therapeutic interventions. Strikingly, lovastatin was found to prevent dendrite degeneration, dopaminergic neuron loss, and impaired locomotion, and, critically, a lovastatin-involved Nrf2 pathway proved neuroprotective (*Lin et al., 2016a*). Nevertheless, whether the Nrf2-mediated neuroprotection is cell- or non-cell-autonomous remains elusive.

Traditional Chinese Medicine (TCM) is often used as an alternative or dietary treatment for human diseases, including PD (*Kim et al., 2012*; *Li et al., 2017*). Although the results were inconclusive, some TCM could display adjuvant effects when used in combination with L-DOPA, reducing the L-DOPA dosage required in long-term treatments and relieving non-motor symptoms (*Kim et al., 2012*). As a prominent component in TCM, *Gastrodia elata* Blume (GE; Orchidaceae) has been used to treat neurological disorders for centuries (*Chen and Sheen, 2011*). GE has been shown to exert neuroprotective, anti-inflammatory, and antioxidative effects in neurodegenerative disease models (*Jang et al., 2015*). The major bioactive compounds in GE include gastrodin and 4-hydroxybenzyl alcohol (4-HBA), both of which display pharmacological effects on neurobiological and psychological disorders (*Chen et al., 2016*; *Kumar et al., 2013*). Additionally, gastrodin and 4-HBA have been reported to activate the Nrf2 signaling in dopaminergic neurons and astrocytes, respectively (*Jiang et al., 2014*; *Luo et al., 2017*), highlighting a potential benefit of incorporating GE in PD treatments. However, the effects and mechanisms underlying how GE moderate Lrrk2-G2019S PD remain unclear.

In the present study, we treated G2019S animals with water extract of GE (WGE), and its bioactive compounds, gastrodin and 4-HBA. We have investigated the impact of WGE treatment on PD in restoring locomotion and protecting dopaminergic neurons in the *Drosophila* G2019S model. We identified two distinct pathways induced by WGE in the model, that is, suppression of Lrrk2 protein accumulation and hyperphosphorylation in neurons, and activation of the Nrf2 pathway in glia, particularly in astrocyte-like and ensheathing glia. We show that WGE-induced Nrf2 activation antagonizes the Gbb-activated Mad signaling in glia, contributing to neuronal protection. WGE also suppressed the hyperactivation of G2019S proteins and antagonized Smad2/3 signaling in a *LRRK2-G2019S* mouse model, which restored locomotion, protected dopaminergic neurons, and regulated the microglia hyperactivation. Conservation of the pathways impacted by WGE treatment in both the *Drosophila* and mouse G2019S models implies the beneficial effects of GE and represent a reliable and effective complementary therapy for PD.

## Results

### WGE treatment improves locomotion of *Ddc>G2019S* flies

We employed the *GAL4-UAS* system to express the human G2019S mutant of Lrrk2 by the *Ddc-GAL4* driver (*Ddc>G2019S*) in dopaminergic neurons and then assessed the anti-geotactic climbing activity of adult flies. We observed that locomotion of *Ddc>G2019S* flies was affected significantly relative to control flies expressing human wild-type Lrrk2 (*Ddc>Lrrk2*) (*Figure 1A*, *Figure 1—figure supplement 1A*). At weeks 1 and 2, more than 80% of *Ddc>G2019S* flies could successfully climb above an 8 cm threshold, a proportion comparable to that of *Ddc>Lrrk2* flies. However, the success rate declined to ~40% at week 3, ~ 20% at week 4, and to less than 10% at weeks 5 and 6. These proportions are significantly lower than the ~80% at week 3, ~60% at week 4, and ~40% at weeks 5 and 6 displayed by *Ddc>Lrrk2* flies. Although both *Ddc>G2019S* and *Ddc>Lrrk2* flies failed to reach the 8 cm mark at weeks 7 and 8, *Ddc>Lrrk2* flies could still climb the wall, whereas almost all

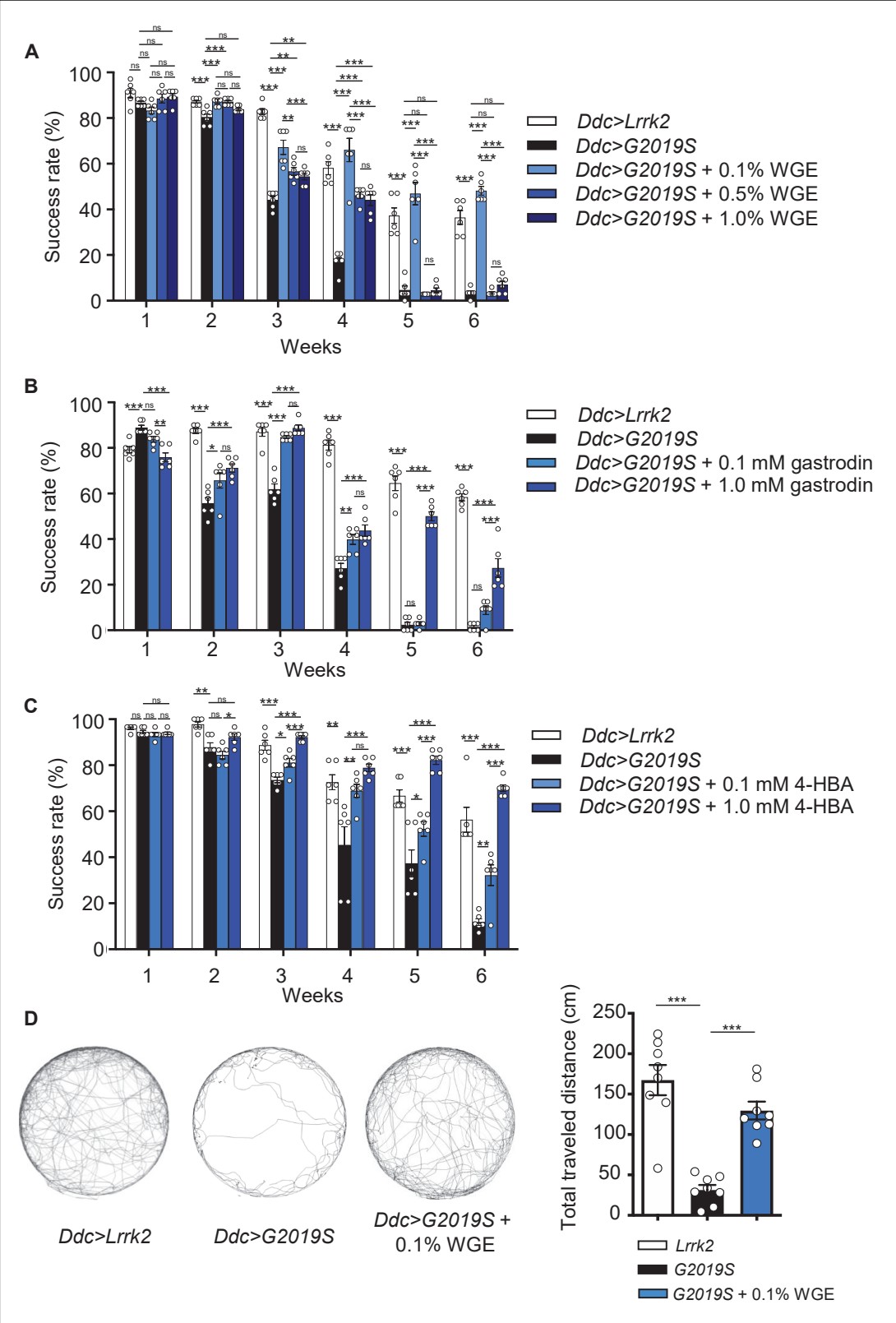

**Figure 1.** Water extract of *Gastrodia elata* Blume (WGE) treatment rescues the diminished locomotion of *Ddc>G2019S* flies. (**A–C**) Climbing activities of *Ddc>G2019S* flies fed on food supplemented with 0.1, 0.5, or 1.0% WGE (**A**), 0.1 or 1.0 mM gastrodin (**B**), and 0.1 or 1.0 mM 4-hydroxybenzyl alcohol (4-HBA) (**C**). Controls are *Ddc>Lrrk2* and *Ddc>G2019S* flies fed regular food. Bar graphs show the percentages of flies (mean ± SEM, N = 6) that climbed above 8 cm within 10 s. (**D**) Five-minute walking tracks pooled from eight flies each of the *Ddc>Lrrk2*, *Ddc>G2019S,* and *Ddc>G2019S* + 0.1% WGE

*Figure 1 continued on next page*

*Figure 1 continued*

groups at week 4. Bar graph at right summarizes their walking distances (mean ± SEM, N = 8). One-way analysis of variance (ANOVA) with Tukey's post-hoc multiple comparison test: *p<0.05, **p<0.01, and ***p<0.001, ns, not significant.

The online version of this article includes the following figure supplement(s) for figure 1:

**Figure supplement 1.** Climbing activity assay of *Ddc>mCD8-GFP* and *Ddc>Lrrk2* flies.

**Figure supplement 2.** Locomotion improvement of *Ddc>G2019S* flies starting water extract of *Gastrodia elata* Blume (WGE) feeding at week 4.

*Ddc>G2019S* flies could not (*Figure 1—figure supplement 1A*). We also tested the climbing activity of another control expressing GFP in dopaminergic neurons (*Ddc>mCD8-GFP*). Both *Ddc>Lrrk2* and *Ddc>mCD8-GFP* flies showed comparable climbing activities in the first six weeks, and a significant number of *Ddc>mCD8-GFP* flies were still able to climb above the 8 cm mark at week 7. Therefore, we used the *Ddc>Lrrk2* line as a control for *Ddc>G2019S* flies in subsequent experiments to dissect the specific mode of pathogenicity of the G2019S mutation.

Next, we examined the effect of feeding flies with WGE as a dietary supplement at different concentrations (0.1, 0.5, or 1.0%, w/w). WGE treatment of *Ddc>G2019S* flies at all three doses elicited a significant improvement in their climbing ability (*Figure 1A*). Strikingly, the lowest concentration (0.1%) of WGE proved the most effective, with *Ddc>G2019S* flies still performing well at climbing (i.e., comparably to *Ddc>Lrrk2* control flies) in weeks 5 and 6. The higher doses of WGE (0.5 and 1.0) still exerted beneficial effects at weeks 3 and 4, albeit not as significantly as the 0.1% dose, but had no beneficial effect in weeks 5 and 6. As 0.1% is the lowest among the three doses tested, we lowered the dose of WGE to 0.02% and found that 0.02% WGE was less effective than 0.1% starting at week 3 till week 6, suggesting that 0.1% is the optimal dose in restoring the climbing activity of *Ddc>G2019S* flies (*Figure 1—figure supplement 1B*). We also tested the WGE effect on the *Ddc>G2019S* flies that were fed with regular food without WGE for 3 weeks. At week 4, these *Ddc>G2019S* flies also showed a significant improvement in their climbing ability compared to the age-matched *Ddc>G2019S* flies fed continuously on regular food (*Figure 1—figure supplement 2*). The effect of improving climbing activity in the WGE-fed *Ddc>G2019S* flies was reduced at week 5 and diminished at week 6, suggesting that WGE feeding starting at earlier stages is important for long-term locomotion improvement.

Gastrodin and 4-HBA are two major phenolic compounds in GE displaying neuropharmacological effects (*Zhan et al., 2016*). Feeding *Ddc>G2019S* flies with food containing gastrodin (0.1 mM) equivalent to the amount in 0.1% WGE also restored locomotion of *Ddc>G2019S* flies in weeks 2–4, though its impact diminished to nonsignificant levels at weeks 5 and 6. However, increasing the gastrodin dose 10-fold (1.0 mM) resulted in improved climbing activity at weeks 5 and 6 (*Figure 1B*). Similarly, the equivalent 0.1 mM of 4-HBA, the aglyconic form of gastrodin and the bioactive form in the brain (*Wu et al., 2017*), was sufficient to restore the climbing ability of *Ddc>G2019S* flies, and a 10-fold dose at 1.0 mM had an even better effect (*Figure 1C*). These results indicate that both gastrodin and 4-HBA are primary bioactive compounds in GE that prevent locomotion decline in *Ddc>G2019S* flies, and higher doses are more beneficial to reach the effect as WGE did.

Success in the antigravity wall-climbing assay also requires an immediate response to startle knockdown. Accordingly, we performed a second assay, free-walking in an open arena, to assess improved locomotion. Consistently, free-walking by *Ddc>G2019S* flies was greatly impaired, with total walking distance reduced to less than 20% that displayed by control *Ddc>Lrrk2* flies (*Figure 1D*). Moreover, *Ddc>G2019S* flies displayed centrophobism, that is, they avoided walking into the central open space. Strikingly, 0.1% WGE feeding greatly extended walking distance and suppressed the centrophobism of *Ddc>G2019S* flies. Together, these two assays strongly indicate that the defective locomotion exhibited by *Ddc>G2019S* flies is greatly improved by feeding them with 0.1% WGE. Because the improvement on the locomotion was more effective in flies fed with 0.1% WGE than the pure compounds, we therefore fed the flies with 0.1% WGE in the following experiments.

## WGE treatment suppresses dopaminergic neuron loss in *Ddc>G2019S* brain

Expression of G2019S mutant protein has been shown to induce a gradual loss of dopaminergic neurons in the adult fly brain, contributing to impaired locomotion (*Lin et al., 2010*; *Liu et al., 2008*).

Several clusters of dopaminergic neurons have been identified in the adult brain of *Drosophila*. Here, we focused on the PPL1, PPL2, PPM1/2, and PPM3 clusters that have well-defined roles in modulating locomotion (*Mao and Davis, 2009*) to assess the effect of WGE treatment. We detected reduced numbers of dopaminergic neurons in the PPL1, PPL2, PPM1/2, and PPM3 clusters of *Ddc>G2019S* flies relative to *Ddc>Lrrk2* controls, which increased in severity from weeks 2 to 6 (*Figure 2*, *Figure 2— figure supplement 1A–C*). Consistently, feeding *Ddc>G2019S* flies with 0.1% WGE restored numbers of dopaminergic neurons in these clusters to the levels observed in controls (*Figure 2*, *Figure 2— figure supplement 1A–C*). Thus, concomitant rescue of locomotion and dopaminergic neuron populations in *Ddc>G2019S* flies indicates that WGE treatment likely promotes dopaminergic neuron survival to restore locomotion.

## WGE treatment suppresses G2019S-induced Lrrk2 hyperactivation

The enhanced survival of dopaminergic neurons due to WGE treatment implies that WGE induces neuroprotective mechanisms in *Ddc>G2019S* flies. The G2019S mutation causes Lrrk2 hyperphosphorylation, protein accumulation, and aberrant cellular signaling (*Price et al., 2018*). Therefore, we explored if WGE-induced neuroprotection is responsible for abrogating these processes in G2019S flies. We detected comparable levels of Lrrk2 proteins in 3-day-old adult brains pan-neuronally expressing wild-type Lrrk2 (*elav>Lrrk2*) or G2019S mutant protein (*elav>G2019S*) (*Figure 3A*). However, phosphorylation levels at the $Ser^{1292}$ autophosphorylation site were higher in the *elav>G2019S* flies compared to *elav>Lrrk2* (*Figure 3A and B*). This outcome was also observed at week 4 (*Figure 3F and G*), consistent with the idea that Lrrk2 is hyperactivated upon G2019S mutation. Hyperphosphorylation in *elav>G2019S* flies led to Lrrk2 protein accumulation, as determined by total Lrrk2 levels at weeks 2 and 4 (*Figure 3D and E*). However, both hyperphosphorylation and protein accumulation were suppressed upon feeding *elav>G2019S* flies with 0.1% WGE (*Figure 3D–G*). In the *elav>Lrrk2* control, WGE feeding had no effect on levels of wild-type Lrrk2 or $Ser^{1292}$ phosphorylation (*Figure 3—figure supplement 1A–C*). We further examined the G2019S mutant-activated downstream effector Rab10, the phosphorylation status of which can serve as an indicator of Lrrk2 kinase activity (*Karayel et al., 2020*). We observed that levels of phosphorylated Rab10 were increased in *elav>G2019S* flies, but this phenotype was suppressed by WGE feeding (*Figure 3F and H*). Hence, feeding flies with WGE suppresses G2019S-induced Lrrk2 protein phosphorylation, accumulation, and signaling.

## WGE treatment restores Akt/GSK3β/Nrf2 pathway activity

The Akt/GSK3β/Nrf2 signaling axis has been shown to promote survival of dopaminergic neurons and ameliorate motor dysfunction in PD models (*Lin et al., 2016b*). We assayed the phosphorylation status of Akt at $Ser^{505}$ in 3-day-old adult fly head extracts and found reduced levels of pAkt in *elav>G2019S* flies relative to *elav>Lrrk2* controls (*Figure 3A and C*). This pAkt reduction persisted at weeks 2 and 4, but was abrogated by feeding *elav>G2019S* flies with 0.1% WGE (*Figure 4A and B*). We also examined phosphorylation levels of Nrf2 at $Ser^{40}$ and GSK3β at $Ser^9$, both of which were reduced upon expression of G2019S mutant protein and were equally offset by WGE feeding (*Figure 4C and D*). Induction of Akt/GSK3β/Nrf2 signaling activates expression of heme oxygenase 1 (HO-1), and we observed diminished levels of this latter protein in *elav>G2019S* flies, which could be rescued by WGE feeding (*Figure 4C and D*). Therefore, WGE feeding restores the Akt/GSK3β/Nrf2 signaling activity compromised by the G2019S mutation.

## Glial Nrf2 mediates the beneficial effect of WGE on restored locomotion and neuronal protection in G2019S flies

Restoration of Akt/GSK3β/Nrf2 signaling activity by WGE treatment prompted us to test by genetic assays if that signaling pathway mediates the WGE mode of action. We focused on the downstream effector Nrf2 encoded by *cap-n-collar* (*cnc*) in *Drosophila*. Intriguingly, neither overexpression (*UAS-cncC-FL2*) nor RNAi knockdown (*UAS-cncTRiP*) of Nrf2 in dopaminergic neurons had an impact on the climbing ability of *Ddc>G2019S* flies (*Figure 5A*). Also, WGE feeding still rescued locomotion of *Ddc>G2019S* flies with Nrf2 knockdown (*Ddc>G2019S; cncTRiP*) to levels comparable to WGE-fed control flies (*Ddc>G2019S; mCD8-GFP*) without Nrf2 knockdown (*Figure 5A*). Thus, the Nrf2 pathway activation by WGE that appears to be effective in protecting neurons and restoring locomotion is likely exerted in cells other than neurons.

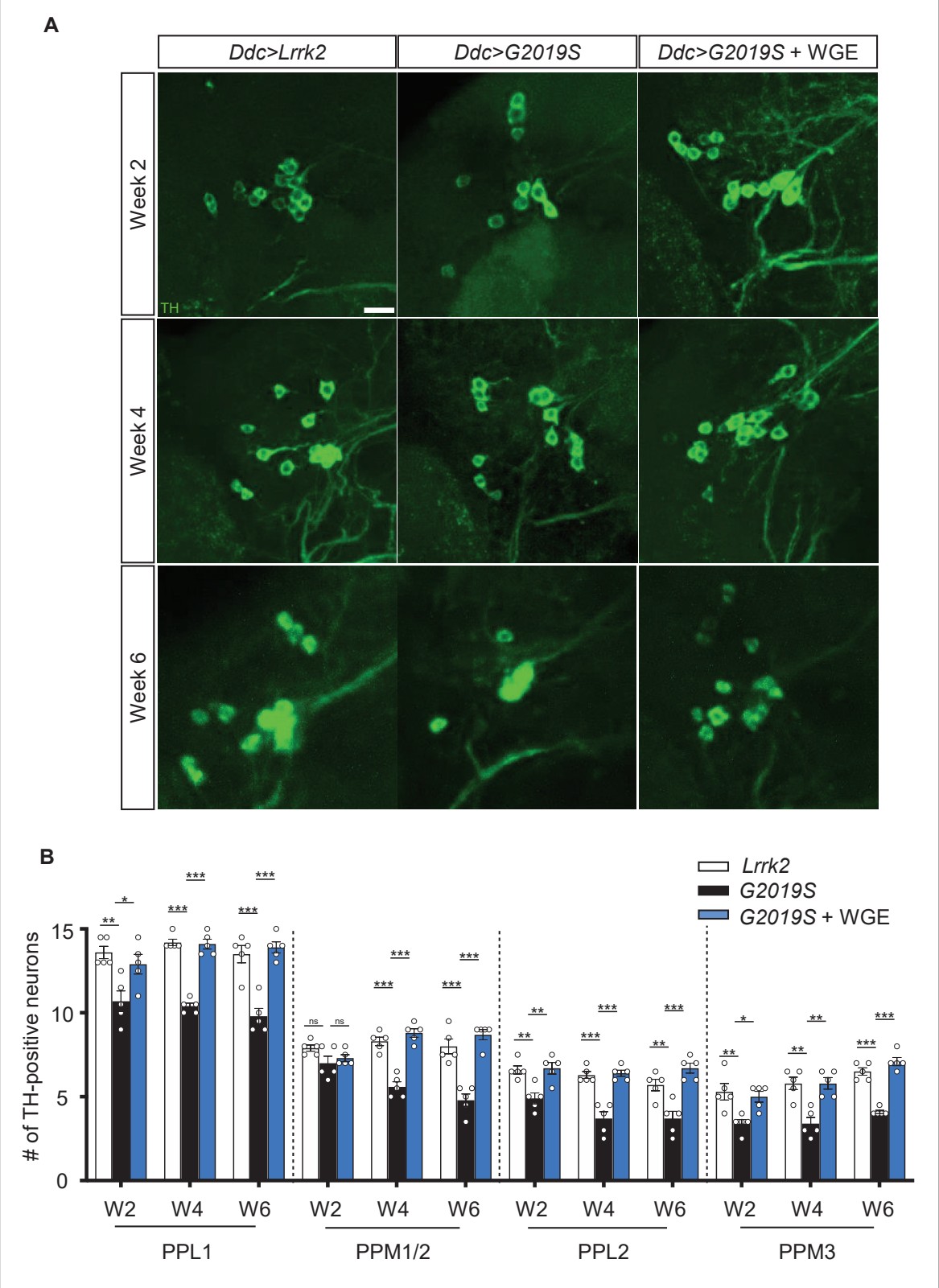

**Figure 2.** Water extract of *Gastrodia elata* Blume (WGE) prevents loss of dopaminergic neurons in *Ddc>G2019S* flies. (**A**) Representative adult brain images showing TH-positive dopaminergic neurons in the PPL1 cluster of 2-, 4-, and 6-week-old flies of the *Ddc>Lrrk2*, *Ddc>G2019S*, and 0.1% WGE-fed *Ddc>G2019S* groups. Scale bar: 10 μm. (**B**) Average numbers (mean ± SEM, N = 5) of TH-positive dopaminergic neurons in the PPL1, PPM1/2, PPL2, and PPL3 clusters per brain hemisphere. One-way ANOVA with Tukey's post-hoc multiple comparison test: *p<0.05, **p<0.01, ***p<0.001, ns, not significant.

*Figure 2 continued on next page*

*Figure 2 continued*

The online version of this article includes the following figure supplement(s) for figure 2:

**Figure supplement 1.** Water extract of *Gastrodia elata* Blume (WGE) treatment prevents dopaminergic neuron loss in *Ddc>G2019S* flies.

Nrf2 activation may be examined by assessing GFP signal from the *ARE-GFP* reporter that harbors Nrf2 binding sites and responds to Nrf2 activation (*Chatterjee and Bohmann, 2012*). *ARE-GFP* adult fly brains displayed low basal GFP signals (*Figure 5—figure supplement 1A*), but feeding with 0.1% WGE elicited many GFP-positive signals in Repo-positive glia (arrowheads in *Figure 5—figure supplement 1B*), evidencing that glia may be the primary cell type in which Nrf2 is activated.

Next, we investigated if WGE-induced Nrf2 activity in glia is effective in promoting locomotion in G2019S flies. To this end, we employed the *LexA-LexAop* system to overexpress wild-type Lrrk2 or G2019S mutant protein in dopaminergic neurons (*Ddc-LexA>Lrrk2* and *Ddc-LexA>G2019S*, respectively) and used the *GAL4* driver to manipulate Nrf2/CncC activity in glia (*repo>cncC-FL2* for overexpression and *repo>cnc-RNAi* for knockdown). As a first step, we validated the age-dependent locomotion decline of *Ddc-LexA>G2019S* flies that was severer than *Ddc-LexA>Lrrk2* and could be rescued by 0.1% WGE feeding (*Figure 5—figure supplement 2*). We then assayed the climbing ability of *Ddc-LexA>G2019S* flies exhibiting *repo-GAL4*-driven Nrf2 overexpression in glia (*Ddc-LexA>G2019S; repo>cncC-FL2*). Significantly, the *Ddc-LexA>G2019S; repo>cncC-FL2* flies performed better in the climbing assay than *Ddc-LexA>G2019S; repo-GAL4* without Nrf2 overexpression (*Figure 5B*). We also performed Nrf2 knockdown in glia (*Ddc-LexA>G2019S; repo>cnc-RNAi*), which had little impact on the already declined climbing activity in *Ddc-LexA>G2019S; repo-GAL4* (*Figure 5B*). Importantly, WGE feeding could not rescue locomotion deficits of glial Nrf2-knockdown flies (*Ddc-LexA>G2019S; repo>cnc-RNAi*) (*Figure 5B*, compare to *Ddc-LexA>G2019S; repo-GAL4* with WGE feeding). Thus, glial overexpression of Nrf2 partially restores locomotion of *Ddc-LexA>G2019S* flies, and glial depletion of Nrf2 abolishes the ability of WGE to rescue impaired locomotion.

We stained dopaminergic neurons of 6-week-old adult fly brains and confirmed that numbers of TH-positive dopaminergic neurons in the PPL1 cluster were reduced in the *Ddc-LexA>G2019S; repo-GAL4* flies compared to *Ddc-LexA>Lrrk2; repo-GAL4* controls (*Figure 5C and D*). Importantly, numbers of dopaminergic neurons in the PPL1 cluster were restored upon glial overexpression of Nrf2 in the *Ddc-LexA>G2019S; repo>cncC-FL2* flies. In contrast, glial Nrf2 knockdown had little impact on the already reduced dopaminergic neurons. Taken together, these results support that glial Nrf2 is compromised in *Ddc-LexA>G2019S* flies, and glial expression of Nrf2 protects the dopaminergic neurons.

Although WGE feeding suppressed hyperactivity of G2019S mutant protein (*Figure 3D–G*), it was not clear if WGE-mediated Nrf2 activation in glia could directly suppress mutant protein hyperactivity in dopaminergic neurons. To test this possibility, we compared the levels of total Lrrk2 protein and phosphorylated Lrrk2 protein in *Ddc-LexA>G2019S* with and without glial Nrf2 overexpression (*Ddc-LexA>G2019S; repo-GAL4* and *Ddc-LexA>G2019S; repo>cncC*). We observed comparable levels of Lrrk2 and pLrrk2 in the control and Nrf2 overexpression lines (*Figure 6—figure supplement 1A–C*). Thus, the neuroprotective effects of Nrf2 activity are unlikely to operate through modulation of Lrrk2 levels or activity.

## Nrf2 in astrocyte-like and ensheathing glia is the major target of WGE

Five types of glia with different morphologies and functions have been identified in the fly brain (*Freeman, 2015*). We decided to identify specific subtypes of glia that may mediate the WGE-induced Nrf2 activity endowing neuronal protection, given that dysfunctional astrocytes and microglia have been linked to onset and progression of both sporadic and familial PD (*Kam et al., 2020*). In *Ddc-LexA>G2019S* flies, *GAL4*-driven *cnc* knockdown in astrocyte-like (*Ddc-LexA>G2019S; alrm>cnc-RNAi*) or ensheathing glia (*Ddc-LexA>G2019S; R56F03>cnc-RNAi*) abolished the improved locomotion elicited by WGE treatment, recapitulating the effect of pan-glial Nrf2 knockdown (*Ddc-LexA>G2019S; repo>cnc-RNAi*) (*Figure 6A*). This outcome was not observed when we used GAL4 drivers to knock down *cnc* in cortex (*np2222*), perineurial (*np6293*), or subperineurial (*moody*) glia. We confirmed the involvement of astrocyte-like and ensheathing glia by means of Nrf2 overexpression in astrocyte-like (*Ddc-LexA>G2019S; alrm>cncC-FL2*) or ensheathing (*Ddc-LexA>G2019S; R56F03>cncC-FL2*) glia, with both treatments improving the climbing activity of *Ddc-LexA>G2019S*

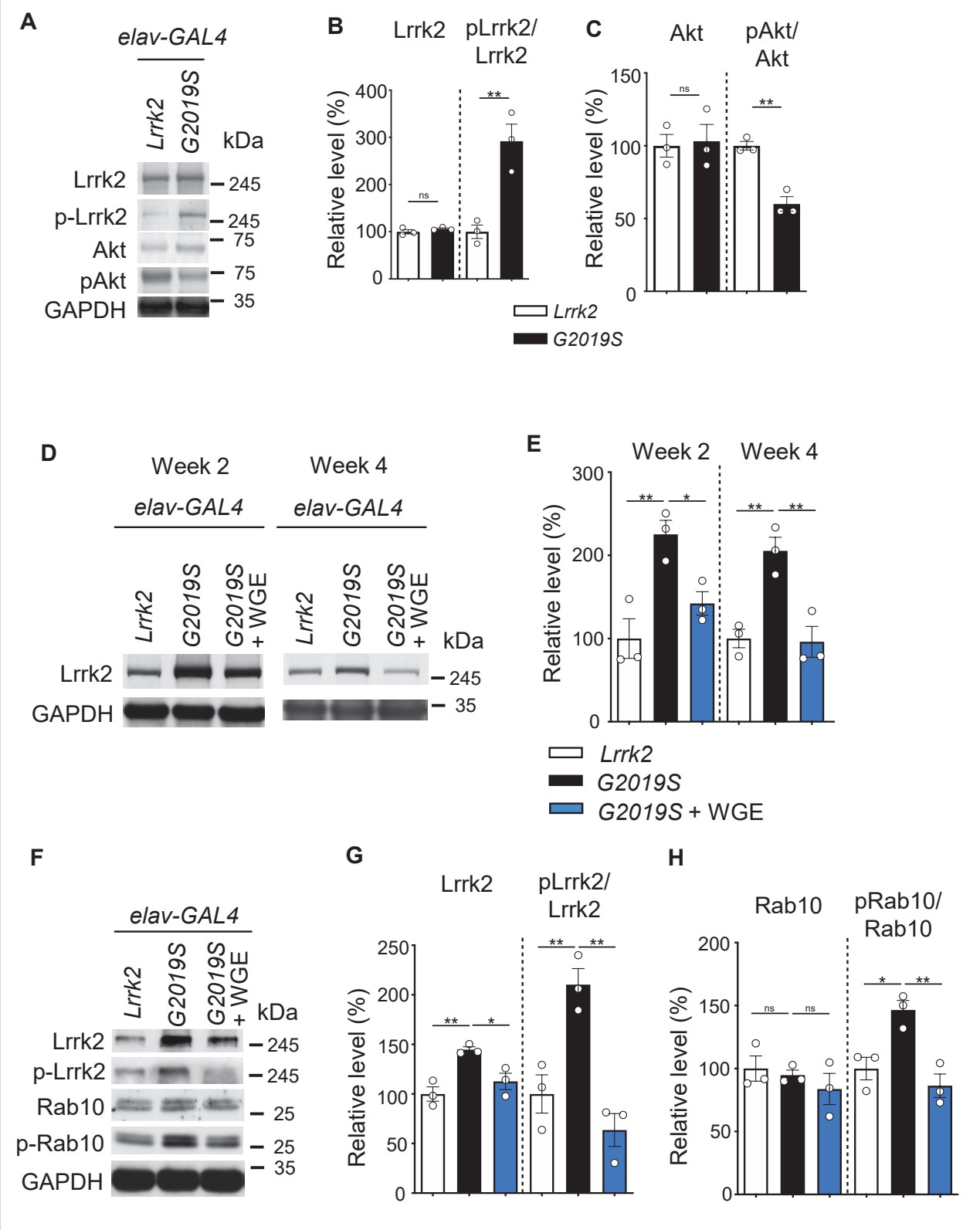

**Figure 3.** Water extract of *Gastrodia elata* Blume (WGE) modulates Lrrk2 accumulation and hyperactivation in *elav>G2019S* flies. (**A**) Representative immunoblots of 3-day-old adult brain lysates showing expression levels of Lrrk2, pLrrk2 (phosphorylated at Ser[1292]), Akt, and pAkt (phosphorylated at Ser[505]) in *elav>Lrrk2* and *elav>G2019S* flies. (**B, C**) Quantifications (mean ± SEM, N = 3) of Lrrk2 and pLrrk2/Lrrk2 (**B**), and Akt and pAkt/Akt (**C**). (**D**) Representative immunoblots of 2- and 4-week-old adult brain lysates showing Lrrk2 levels in the fly groups *elav>Lrrk2*, *elav>G2019S*, and *elav>G2019S*

*Figure 3 continued on next page*

*Figure 3 continued*

fed with 0.1% WGE. (**E**) Quantification of Lrrk2 levels in 2- and 4-week-old adult brains (mean ± SEM, N = 3). (**F**) Representative immunoblots of 4-week-old adult brain lysates showing expression levels of Lrrk2, pLrrk2, Rab10, and pRab10 (phosphorylated at Thr[73]). (**G, H**) Quantification (mean ± SEM, N = 3) of levels of Lrrk2 and pLrrk2/Lrrk2 (**G**), and of Rab10 and pRab10/Rab10 (**H**). One-way ANOVA with Tukey's post-hoc multiple comparison test: *$p<0.05$, **$p<0.01$, ns, not significant.

The online version of this article includes the following figure supplement(s) for figure 3:

**Figure supplement 1.** Water extract of *Gastrodia elata* Blume (WGE) specifically modulates Lrrk2 accumulation and hyperactivation in *elav>G2019S* but not *elav> Lrrk2* flies.

flies not subjected to WGE feeding (***Figure 6B***). These analyses indicate that WGE feeding induces Nrf2 activity in astrocyte-like and ensheathing glia, which mitigates the reduced locomotion displayed by G2019S mutant-expressing flies.

Next, we assayed Nrf2-regulated *ARE-GFP* expression in astrocyte-like and ensheathing glia of flies expressing G2019S mutant protein and subjected to WGE treatment. We focused on the astrocyte-like and ensheathing cells located adjacent to dopaminergic neurons. In control flies expressing wild-type Lrrk2, we detected basal levels of GFP signal in mCherry-positive astrocyte-like (*alrm-GAL4*) or ensheathing glia (*R56F03*) (***Figure 6C–F***). We detected lower levels of GFP signal in these cells when G2019S mutant protein was expressed in dopaminergic neurons. However, upon feeding with 0.1% WGE, we observed higher levels of GFP signal in both types of glia. Changes in the intensities of GFP signals were also detected in TH-positive dopaminergic neurons, although the levels were lower than in glia (***Figure 6—figure supplement 2A and B***). Thus, overexpression of G2019S mutant protein in dopaminergic neurons elicited reduced Nrf2 signaling activity, but WGE feeding restored or further enhanced Nrf2 activities in these two glial subtypes.

## Nrf2 activation antagonizes BMP signaling in glia

G2019S mutant protein in dopaminergic neurons has been shown previously to enhance the expression of the proprotein convertase Furin 1 (Fur1) that processes the BMP signaling molecule Gbb for maturation and release, leading to activation of the BMP signaling pathway in glia (***Maksoud et al., 2019***). We confirmed that finding by removing one copy of *Mad* that encodes the pathway's downstream effector to restore locomotion in *Ddc>G2019S* flies (***Figure 7—figure supplement 1***). To address if WGE-induced Nrf2 activation could antagonize BMP signaling activity in glia, first we assessed expression of the phosphorylated Mad (pMad) activated by BMP signaling. In glia of *Ddc>G2019S* adult fly brains, pMad levels were higher than in *Ddc>Lrrk2* brains and they could be suppressed by WGE treatment (***Figure 7A and B***). Similar to the previous report (***Maksoud et al., 2019***), glial overexpression of Mad (*UAS-Mad*) or constitutively active type I receptor Tkv (*UAS-tkv*^Q253D^) was sufficient to impair locomotion, even without expressing G2019S mutant protein in neurons. However, impaired locomotion was rescued in both cases by 0.1% WGE feeding (***Figure 7C and D***). In addition, numbers of dopaminergic neurons in the PPL1 cluster were reduced upon glial overexpression of *Mad* or *tkv*^Q253D^ and they were restored by 0.1% WGE treatment (***Figure 7E–H***). Together, these analyses indicate that activation of BMP signaling in glia recapitulates the phenotypes observed in flies over-expressing G2019S mutant protein in dopaminergic neurons and, furthermore, that these effects can be suppressed by WGE treatment.

Next, we explored if WGE-induced Nrf2 activation antagonizes BMP signaling in glia. Glial Nrf2 overexpression in *Ddc-LexA>G2019S* flies (*Ddc-LexA>G2019S; repo>cncC-FL2*) partially rescued the locomotory impairment caused by G2019S mutation (***Figure 5B***), and removing one copy of *Mad* (*Ddc-LexA>G2019S, Mad^{+/-}; repo>cncC-FL2*) further enhanced this effect (***Figure 8A***). Indeed, this outcome was equivalent to removing one copy of *Mad* but without Nrf2 overexpression (*Ddc-LexA>G2019S, Mad^{+/-}; repo-GAL4*), suggesting that Nrf2 functions mainly to antagonize Mad activity (***Figure 8A***). Consistently, Nrf2 overexpression in glia or WGE treatment suppressed the upregulation of pMad levels in glia caused by G2019S mutant protein overexpression in dopaminergic neurons (*Ddc-LexA>G2019S; repo-GAL4*) (***Figure 8B and C***). Depletion of Nrf2 from glia (*Ddc-LexA>G2019S; repo>cnc-RNAi*), even in the presence of WGE treatment, maintained high pMad levels in glia (***Figure 8B and C***). Thus, modulation of Nrf2 activity, either by genetic manipulation or by WGE treatment, has an antagonistic effect on pMad levels in glia.

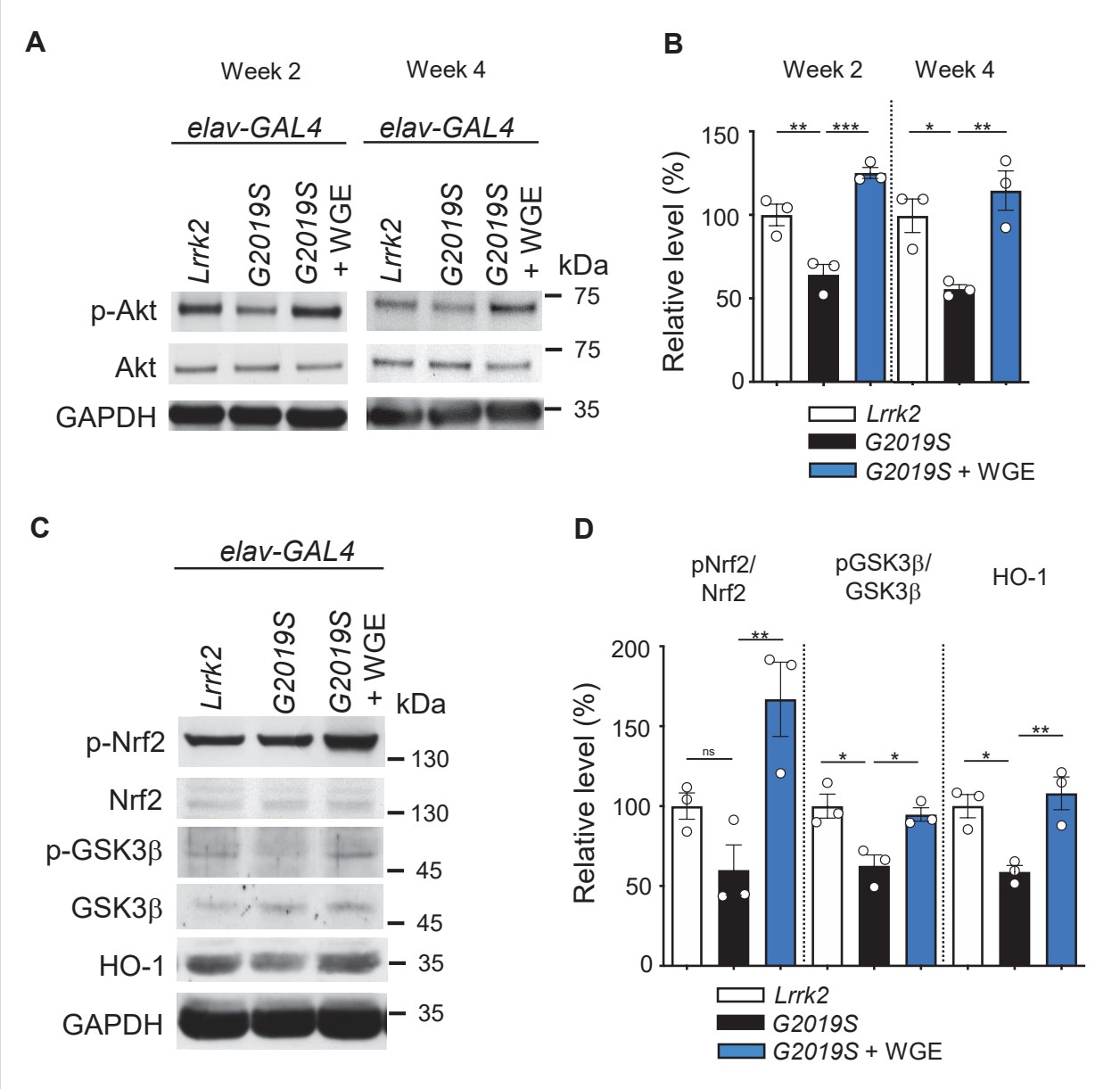

**Figure 4.** Water extract of *Gastrodia elata* Blume (WGE) activates the Akt-Nrf2 pathway in *elav>G2019S* flies. (**A**) Representative immunoblots of 2- and 4-week-old adult brain lysates showing levels of Akt and pAkt in brain extracts of *elav>Lrrk2*, *elav>G2019S*, and WGE-fed *elav>G2019S* flies. (**B**) Quantification (mean ± SEM, N = 3) of pAkt/Akt levels in 2- and 4-week-old adult brains. (**C**) Representative immunoblots of 4-week-old adult brain lysates showing levels of Nrf2, pNrf2 (phosphorylated at Ser[40]), GSK3β, pGSK3β (phosphorylated at Ser[9]), and HO-1 in *elav>Lrrk2*, *elav>G2019S*, and WGE-fed *elav>G2019S* flies. GADPH acted as a loading control in (**A**) and (**C**). (**D**) Quantification (mean ± SEM, N = 3) of relative protein levels to respective Nrf2, GSK3β, and HO-1. One-way ANOVA with Tukey's post-hoc multiple comparison test (relative to *elav>G2019S*): *p<0.05, **p<0.01, ***p<0.001, ns, not significant.

We also addressed if *Mad* modulates expression of the Nrf2 target *ARE-GFP*. We observed diminished GFP signal in PPL1-surrounding glia of *Ddc-LexA>G2019S* adult fly brains relative to the *Ddc-LexA>Lrrk2* control. Heterozygosity of *Mad* in *Ddc-LexA>G2019S* flies (*Ddc-LexA>G2019S, Mad+/-*) restored the level of GFP signal, implying that Mad modulates targeted gene expression induced by Nrf2 activity (*Figure 8D and E*).

## The effects of WGE treatment in a *LRRK2-G2019S* mouse model

To further study the effect of WGE on G2019S mutation-induced neurodegeneration in a mammalian system, we fed *LRRK2-G2019S* transgenic mice with WGE starting at the age of 8.5 months, that

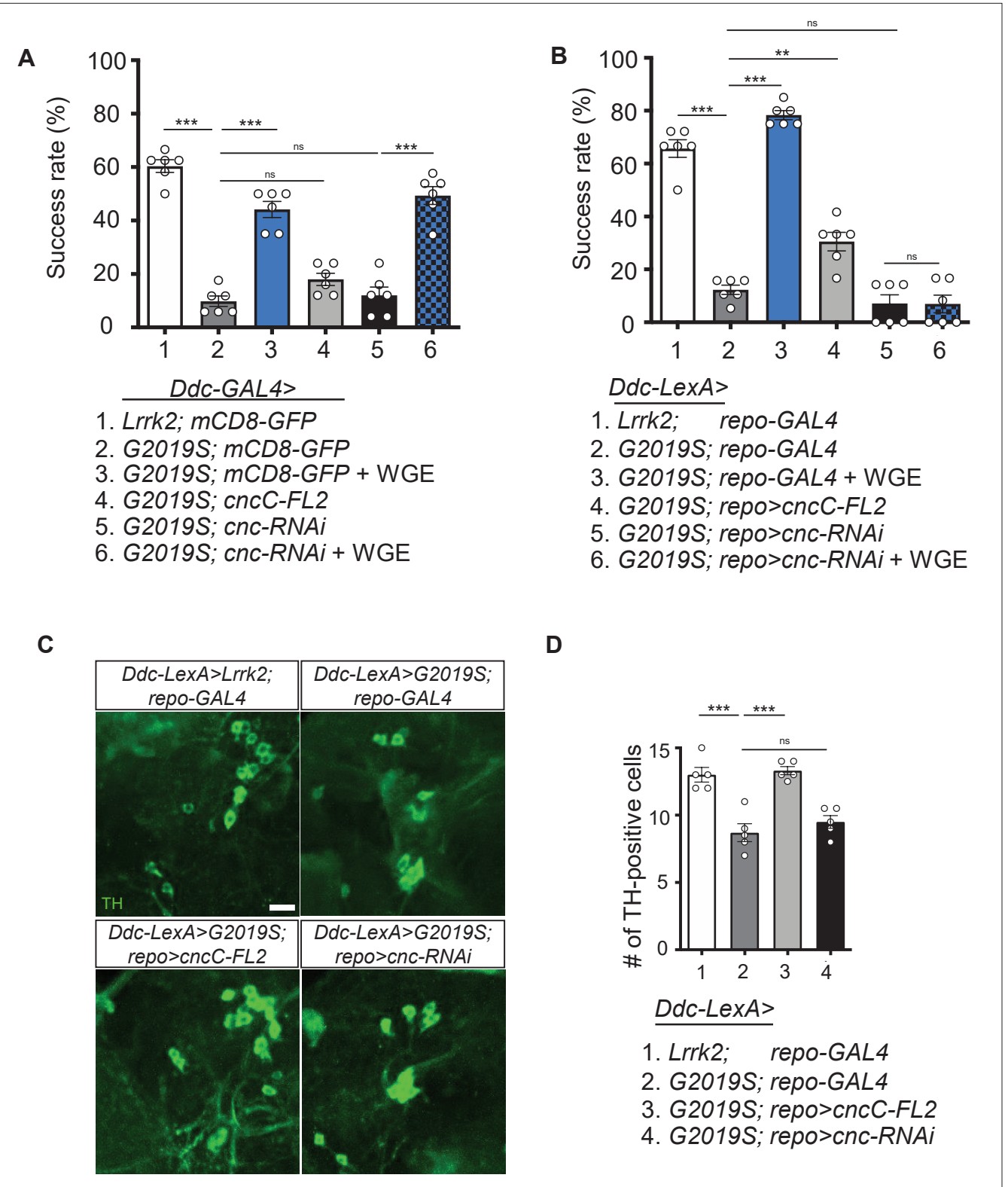

**Figure 5.** Activation of Nrf2 in glia rescues locomotion defects in *Ddc>G2019S* flies. (**A, B**) Requirement of Nrf2 in glia but not neurons for water extract of *Gastrodia elata* Blume (WGE)-improved *Ddc>G2019S* climbing activity. (**A**) Climbing success rates of flies in which *Ddc-GAL4* drives coexpression of *Lrrk2-G2019S* and *mCD8-GFP*, *cncC-FL2* or *cnc-RNAi*. As a control line, *Ddc-GAL4* drives coexpression of *Lrrk2* and *mCD8-GFP*. (**B**) Climbing success rates of flies in which *Ddc-LexA* drives wild-type *Lrrk2* or *Lrrk2-G2019S* expression and *repo-GAL4* drives *cncC-FL2* or *cnc-RNAi* expression (mean ±

*Figure 5 continued on next page*

*Figure 5 continued*

SEM, N = 6 for **A** and **B**). WGE was added to food at a concentration of 0.1% (w/w). (**C**) Adult brain images showing TH-positive dopaminergic neurons in the PPL1 clusters of 6-week-old *Ddc-LexA>Lrrk2* or *Ddc-LexA>G2019S* flies with *repo-GAL4* control or *repo-GAL4*-driven *cncC-FL2* or *cnc-RNAi* expression. Scale bar: 10 μm. (**D**) Average numbers (mean ± SEM, N = 5) of TH-positive dopaminergic neurons in PPL1 clusters per brain hemisphere are shown. One-way ANOVA with Tukey's post-hoc multiple comparison test (relative to *Ddc>G2019S* [**A**] or *Ddc-LexA>G2019S; repo-GAL4* [**B, D**]): **p<0.01, ***p<0.001, ns, not significant.

The online version of this article includes the following figure supplement(s) for figure 5:

**Figure supplement 1.** Water extract of *Gastrodia elata* Blume (WGE) treatment specifically activates glial Nrf2 signals in the *ARE-GFP* reporter flies.

**Figure supplement 2.** Water extract of *Gastrodia elata* Blume (WGE) rescues the locomotion defect displayed by *Ddc-LexA>G2019S* flies.

is, prior to onset of impairments in locomotion and dopaminergic neurons (*Chou et al., 2014*). We quantified three locomotor activities from video-tracking paths of an open-field test, that is, accumulative moving distance, average velocity, and percentage of time moving (*Figure 9A and B*). At the age of 8.5 months, three groups—non-transgenic (nTg), transgenic *LRRK2-G2019S*, and WGE-fed transgenic *LRRK2-G2019S* mice—presented comparable locomotor activities. At 9.5 months of age, the *LRRK2-G2019S* mice displayed clearly impaired locomotion, which was statistically significant at the age of 11.5 months relative to nTg littermates (*Figure 9B*), consistent with a previous report (*Chen et al., 2012*). Importantly, the *LRRK2-G2019S* mice fed with WGE showed improved locomotion throughout the 3-month treatment period, an outcome that was statistically significant at 11.5 months (*Figure 9A and B*). WGE treatment also suppressed the centrophobism displayed by *LRRK2-G2019S* mice (*Figure 9A*). We also analyzed the gait of these three groups of mice (*Figure 9C*). Similar to our findings from the open-field test, stride length of *LRRK2-G2019S* mice was significantly reduced at 11.5 months, but it was restored to the level of nTg mice by WGE feeding (*Figure 9C and D*). Collectively, these analyses show that WGE feeding is an effective means of restoring G2019S mutation-induced locomotory declines in this mouse model of PD.

Since the nigral-striatal system contributes to locomotor function, we counted the number of TH-positive dopaminergic neurons in the substantia nigra. In comparison to nTg littermates, the number of dopaminergic neurons in 11.5-month-old *LRRK2-G2019S* mice was significantly reduced, but WGE treatment for 3 months abrogated this loss of dopaminergic neurons (*Figure 10A and B*). In the *Ddc>G2019S Drosophila* model, glia mediate the protective effects of WGE on dopaminergic neurons. We found that activated microglia marked by ionized calcium-binding adapter molecule 1 (Iba-1) were increased in the substantia nigra of *LRRK2-G2019S* mice, but this increase was suppressed by WGE treatment (*Figure 10C and D*, *Figure 10—figure supplement 1*). We further analyzed levels of activated LRRK2 (pLRRK2) in lysates isolated from the nigra striatum. Levels of pLRRK2 normalized to LRRK2 signal were higher in *LRRK2-G2019S* mice than in nTg littermates, but WGE treatment partially abrogated that outcome (*Figure 10E and F*). Since G2019S mutation enhanced Mad signaling activation in the fly brain (*Figure 8B and C*), we also tested this scenario in the mouse model. Immunoblots revealed that the ratio of phosphorylated Smad2 to total Smad2 (pSmad2/Smad2) was significantly elevated in *LRRK2-G2019S* mice, but this increase was suppressed by WGE treatment (*Figure 10E and G*). Moreover, although the level of pSmad3/Smad3 was not significantly enhanced by G2019S mutant protein expression, it was suppressed by WGE treatment (*Figure 10E and G*). Thus, WGE feeding suppresses G2019S mutation-induced microglia activation and Smad signaling in the substantia nigra of the *LRRK2-G2019S* mouse model of PD.

In summary, the potential mechanisms of WGE involved in the modulation of the neuron-glial interaction are proposed (*Figure 11*). WGE regulates the hyperactivation of G2019S mutant protein in dopaminergic neurons and antagonizes the Mad signaling by activating the Nrf2 pathway in glia. Both actions provide neuroprotection.

## Discussion

Motor dysfunction in *Drosophila* neurodegeneration models has been frequently evaluated by means of negative-geotaxis assay that measures the insect innate response. The assay begins with a sudden external stimulation, which initially inhibits spontaneous locomotion but is followed by climbing behavior. The entire response requires motor circuit coordination and muscle tone regulation, two processes that progressively decline with age and that are impacted by neuropathological insults

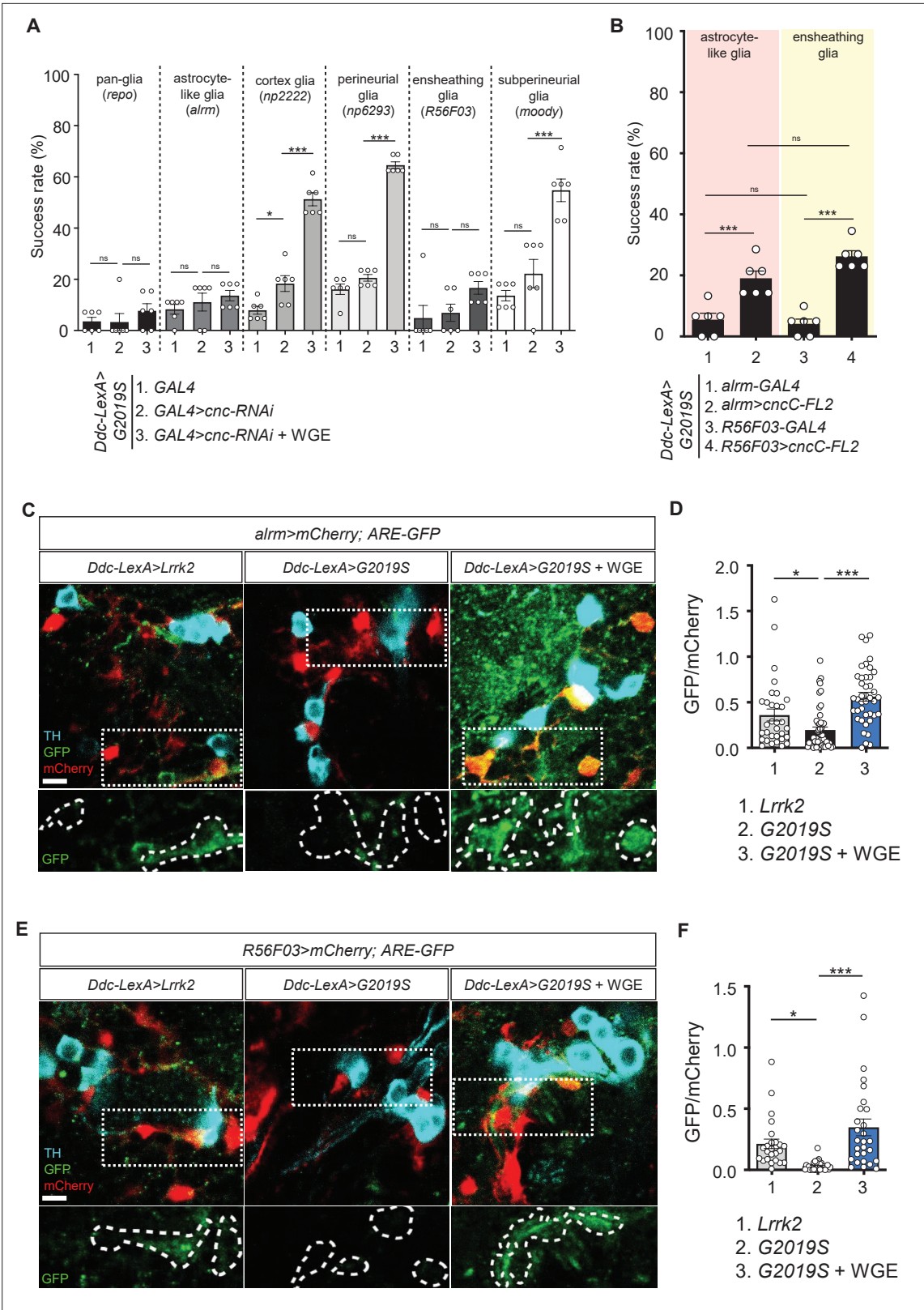

**Figure 6.** Nrf2 in astrocyte-like and ensheathing glia mediates the effect of water extract of *Gastrodia elata* Blume (WGE) treatment in *Ddc>G2019S* flies. (**A**) Nrf2 knockdown in astrocyte-like and ensheathing glia abolishes the improved locomotion elicited by WGE treatment in *Ddc>G2019S* flies. Composite bar graph shows climbing success rates for 6-week-old *Ddc-LexA>G2019S* flies with *cnc-RNAi* driven by *repo-GAL4* in all glia, *alrm-GAL4* in astrocyte-like glia, *np2222* in cortex glia, *np6293* in perineurial glia, *R56F03* in ensheathing glia, and *moody-GAL4* in subperineurial glia. (**B**) Composite

*Figure 6 continued on next page*

*Figure 6 continued*

bar graph shows climbing success rates (mean ± SEM, N = 6) for 6-week-old *Ddc-LexA>G2019S* flies with overexpression of Nrf2 in astrocyte-like glia (*alrm>cncC-FL2*) or ensheathing glia (*R56F03>cncC-FL2*). One-way ANOVA and Tukey's post-hoc multiple comparison test (relative to *Ddc-LexA>G2019S; GAL4>cnc-RNAi* [**A**] or *Ddc-LexA>G2019S; GAL4* [**B**]): *p<0.05, ***p<0.001, ns, not significant. (**C, E**) Representative images showing expression of *ARE-GFP* in astrocyte-like glia (*alrm>mCherry*) (**C**) and ensheathing glia (*R56F03>mCherry*) (**E**), together with TH-positive dopaminergic neurons in the PPL1 clusters of 6-week-old *Ddc-LexA>Lrrk2*, *Ddc-LexA>G2019S*, or WGE (0.1% w/w)-fed *Ddc-LexA>G2019S* flies. Bar: 5 µm. GFP channels in the dashed boxes are shown as enhanced views in the lower panel, with glial signals labeled by dashed lines. (**D, F**) Quantifications (mean ± SEM, n > 25 for each genotype) of GFP intensities in astrocyte-like glia (**D**) or ensheathing glia (**F**). GFP intensities in glia have been outlined manually using the mCherry-positive signals. One-way ANOVA and Tukey's post-hoc multiple comparison test (relative to *Ddc-LexA>G2019S; alrm>mCherry* [**D**] or *Ddc-LexA>G2019S; R56F03>mCherry* [**F**]): *p<0.05, ***p<0.001, ns, not significant.

The online version of this article includes the following figure supplement(s) for figure 6:

**Figure supplement 1.** Lrrk2 and pLrrk2 levels are maintained upon glial Nrf2 overexpression.

**Figure supplement 2.** Water extract of *Gastrodia elata* Blume (WGE) treatment induces a mild Nrf2 activation in dopaminergic neurons in *Ddc>G2019S* flies.

---

(*Grotewiel et al., 2005*). In contrast, the open-arena walking assay allows free exploration without disturbance, representing an assay for locomotor activities that can reveal deficits like bradykinesia (*Chen et al., 2014*). Our G2019S flies exhibited reduced locomotor activity in both assays. Moreover, the G2019S flies also exhibited centrophobism-like behavior in the open arena (*Figure 1D*). Centrophobism is indicative of emotional abnormalities, such as anxiety and depression, both of which are often associated with PD (*Kulisevsky et al., 2008*), and was also displayed by the *LRRK2-G2019S* mice (*Figure 9A*). WGE treatment suppresses centrophobism in both fly and mouse PD models and has important implications for tackling major symptoms of PD and even non-PD-related depression, as reported for rodent models (*Lin et al., 2018*; *Lin et al., 2016b*). Thus, WGE treatment exerts beneficial effects in both of our PD models.

We found that the 0.1% dosage of WGE is optimal in suppressing age-dependent locomotion decline in G2019S flies, with higher and lower doses being less effective (*Figure 1A*, *Figure 1—figure supplement 1B*). Although an inverted U-shaped drug response is common, a plausible explanation for the diminished effectiveness of higher doses is that WGE downregulates the day-time but not night-time locomotor activity of flies (*Jo et al., 2017*). In mice, higher WGE doses have sleep-promoting effects by activating adenosine $A_1/A_2A$ receptors in the ventrolateral preoptic area (*Zhang et al., 2012*). Thus, dosage level is critical to the beneficial effects of GE in both PD models.

Gastrodin and 4-HBA are considered the principal active components in GE (*Zhan et al., 2016*). Although both compounds can cross the blood-brain barrier (*Wu et al., 2017*), the capability of gastrodin to do so is relatively poor compared to aglyconic 4-HBA due to the glucose moiety (*Lin et al., 2007*). Gastrodin is quickly metabolized to 4-HBA and undetermined metabolites in the brain (*Lin et al., 2008*), perhaps explaining the lower effectiveness of gastrodin in G2019S flies. However, it is likely that the combination of gastrodin with other components in GE might exert optimal beneficial effects.

Expression of G2019S mutant protein impacts different clusters of dopaminergic neurons in the fly brain that are known for their connectivity and function. Activation of two specific mushroom body (MB)-projection dopaminergic neurons in the PPL1 cluster inhibits climbing performance (*Sun et al., 2018*). Mutations in the circadian gene *Clock* (*Clk*) cause PPL1 dopaminergic neuron degeneration, accelerating impaired age-associated climbing ability (*Vaccaro et al., 2017*). Dopaminergic neurons in the PPL2 cluster extend processes to the calyx of the MB, which has been linked to climbing activity (*Sun et al., 2018*). In contrast, PPM3 neurons project to the central complex, activation of which enhances locomotion (*Kong et al., 2010*). A significant reduction in dopaminergic neurons in the PPM1/2 cluster was only found at week 4 when impaired locomotion of G2019S flies was prominent. In a PD fly model involving *SNCA* overexpression and *aux* knockdown, dopaminergic neurons in the PPM1/2 cluster were selectively degenerated and this phenotype was accompanied by impaired locomotion in relatively young adult flies (*Song et al., 2017*). We postulate that the age-dependent impaired locomotion displayed by G2019S flies could be caused by gradual and differential loss of dopaminergic neurons in these clusters, thereby affecting different aspects of locomotion. However, further study is needed to test that hypothesis.

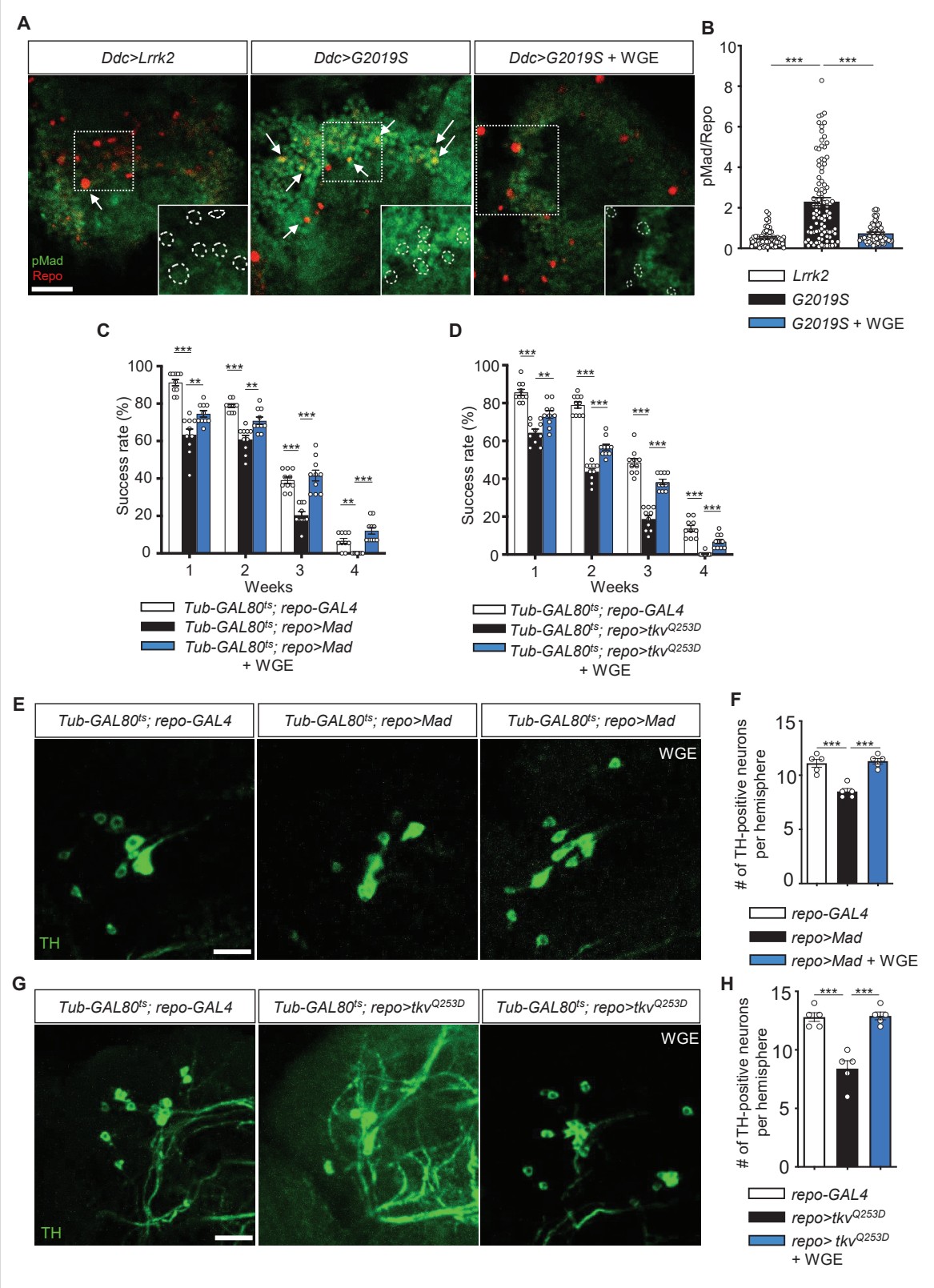

**Figure 7.** Water extract of *Gastrodia elata* Blume (WGE) downregulates G2019S-induced BMP/Mad signaling. (**A**) Representative images of glial pMad staining in the adult PPL1 clusters of *Ddc>Lrrk2*, *Ddc>G2019S*, and WGE-fed *Ddc>G2019S* flies. White arrows indicate pMad signals co-localized with Repo, with single channels for pMad signals shown as insets. Bar: 10 μm. (**B**) Quantification (mean ± SEM, n > 60 for each genotype) of pMad signals normalized to Repo levels in glia of the indicated genotypes. One-way ANOVA and Tukey's post-hoc multiple comparison test (relative to *Ddc>G2019S*):

*Figure 7 continued on next page*

*Figure 7 continued*

***p<0.001. (**C, D**) Climbing success rates (mean ± SEM, N = 10) at weeks 1–4 demonstrating that WGE treatment rescues locomotion deficits induced by glial overexpression of *Mad* (**C**) or *tkv*$^{Q253D}$ (**D**) in *Tub-GAL80*$^{ts}$; *repo-GAL4* flies. One-way ANOVA and Tukey's post-hoc multiple comparison test (relative to *repo>Mad* or *repo>tkvQ*$^{Q253D}$): **p<0.01, ***p<0.001. (**E, G**) Representative images of 4-week-old adult brain showing TH staining of the PPL1 clusters of *Tub-GAL80*$^{ts}$; *repo-GAL4*, *Tub-GAL80*$^{ts}$; *repo>Mad*, and WGE-fed *Tub-GAL80*$^{ts}$; *repo>Mad* flies (**E**), and *Tub-GAL80*$^{ts}$; *repo-GAL4*, *Tub-GAL80*$^{ts}$; *repo>tkvQ*$^{Q253D}$, and WGE-fed *Tub-GAL80*$^{ts}$; *repo>tkvQ*$^{Q253D}$ flies (**G**). Bars: 12.5 μm. (**F, H**) Bar graphs show mean ± SEM (N = 5) of TH-positive dopaminergic neurons in the PPL1 clusters of 4-week-old flies. One-way ANOVA and Tukey's post-hoc multiple comparison test (relative to *Tub-GAL80*$^{ts}$; *repo>Mad* [**F**] or *Tub-GAL80*$^{ts}$; *repo>tkvQ*$^{Q253D}$ [**H**]): ***p<0.001.

The online version of this article includes the following figure supplement(s) for figure 7:

**Figure supplement 1.** *Mad* heterozygosity rescues the impaired locomotion of *Ddc>G2019S* flies.

Expression of the G2019S mutant protein induces Lrrk2 auto- and hyperphosphorylation, as well as protein accumulation, together enhancing cellular Lrrk2 activity and causing aberrant downstream signaling (*Sheng et al., 2012*). We have shown here that neuronal expression of Lrrk2-G2019S reduced Akt phosphorylation (*Figure 4A and B*). Consistently, hyperactivated G2019S mutant protein impaired interaction with and phosphorylation of Akt, resulting in compromised signaling and accelerated neurodegeneration (*Ohta et al., 2011*; *Panagiotakopoulou et al., 2020*). However, WGE feeding restored downstream Akt signaling by suppressing G2019S mutant protein hyperactivation. Rab10, one of the best-characterized substrates for Lrrk2, mediates several of Lrrk2's cellular functions (*Karayel et al., 2020*). In *Drosophila*, Rab10 and Lrrk2-G2019S synergistically affect the activity of dopaminergic neurons, mediating deficits in movement (*Fellgett et al., 2021*; *Petridi et al., 2020*). We have shown that WGE treatment downregulated levels of Lrrk2 accumulation and phosphorylated Rab10 (*Figure 3F–H*), thus alleviating their synergistic toxicity. Several kinase inhibitors have been developed to block the kinase activity of Lrrk2, including of both wild-type Lrrk2 and the G2019S mutant, which could affect endogenous Lrrk2 activity (*Sheng et al., 2012*). Instead, WGE treatment modulates the phosphorylation status and protein level of the G2019S mutant but not those of wild-type Lrrk2. The new hydrogen bond created at the Ser$^{2019}$ autophosphorylation site may provide a docking site for some chemicals in WGE, representing a possible explanatory mechanism that warrants further study (*Lang et al., 2015*).

The antioxidation and detoxification factor Nrf2 is a target of Akt activation. Nrf2 phosphorylation and HO-1 expression levels revealed that Nrf2 is inactivated in G2019S flies, but it was activated by WGE treatment (*Figure 4C and D*). Intriguingly, our genetic data indicate that Nrf2 primarily functions in the glia of G2019S flies, with Nrf2 depletion from glia eliminating the beneficial effects of WGE and glial Nrf2 activation partially substituting for WGE feeding (*Figure 5B*). Cortical neurons express much lower levels of Nrf2 than astrocytes owing to hypo-acetylation and transcriptional repression of the *Nrf2* promoter (*Bell et al., 2015*). Moreover, neurons express greater amounts of Cullin 3, the scaffold component of the E3 ubiquitin ligase that targets Nrf2 for proteasomal degradation (*Jimenez-Blasco et al., 2015*). Both those mechanisms render neuronal Nrf2 inert to activation. Nrf2 activation in astrocytes maintains neuronal integrity and function against oxidative insults in response to stress by supplying antioxidants such as glutathione and HO-1 (*Kraft et al., 2004*; *Vargas and Johnson, 2009*). Previous study showed that 4-HBA triggers glia to secrete HO-1 via the Nrf2 pathway, protecting neurons from hydrogen peroxide in the primary culture (*Luo et al., 2017*). In PD models in which wild-type or mutant α-synuclein is overexpressed, activation of neuronal Nrf2 (*Barone et al., 2011*; *Skibinski et al., 2017*) or astrocytic Nrf2 (*Gan et al., 2012*) proved neuroprotective. In a previous study, lovastatin treatment provides neuroprotection in the G2019S-induced PD model, also through the Akt/Nrf2 pathway (*Lin et al., 2016b*). As activation of neuronal Nrf2 plays a nonconventional role in promoting developmental dendrite pruning (*Chew et al., 2021*), it remains interesting to further study the cell types that mediate the action of lovastatin. By genetically manipulating the G2019S fly model, we have shown that WGE-induced Nrf2 activation in glia but not in neurons protects dopaminergic neurons from degeneration and ameliorates impaired locomotion.

Astrocyte-like and ensheathing glia are two major types of glia in the *Drosophila* nervous system, surrounding and also extending long processes into neuropils of the brain. These astrocyte-like glia exhibit a morphology and function similar to those of mammalian astrocytes, including reuptake of neurotransmitters and phagocytosis of neuronal debris (*Freeman, 2015*; *Tasdemir-Yilmaz and Freeman, 2014*). Ensheathing glia of varying morphologies encase axonal tracts and neuropils,

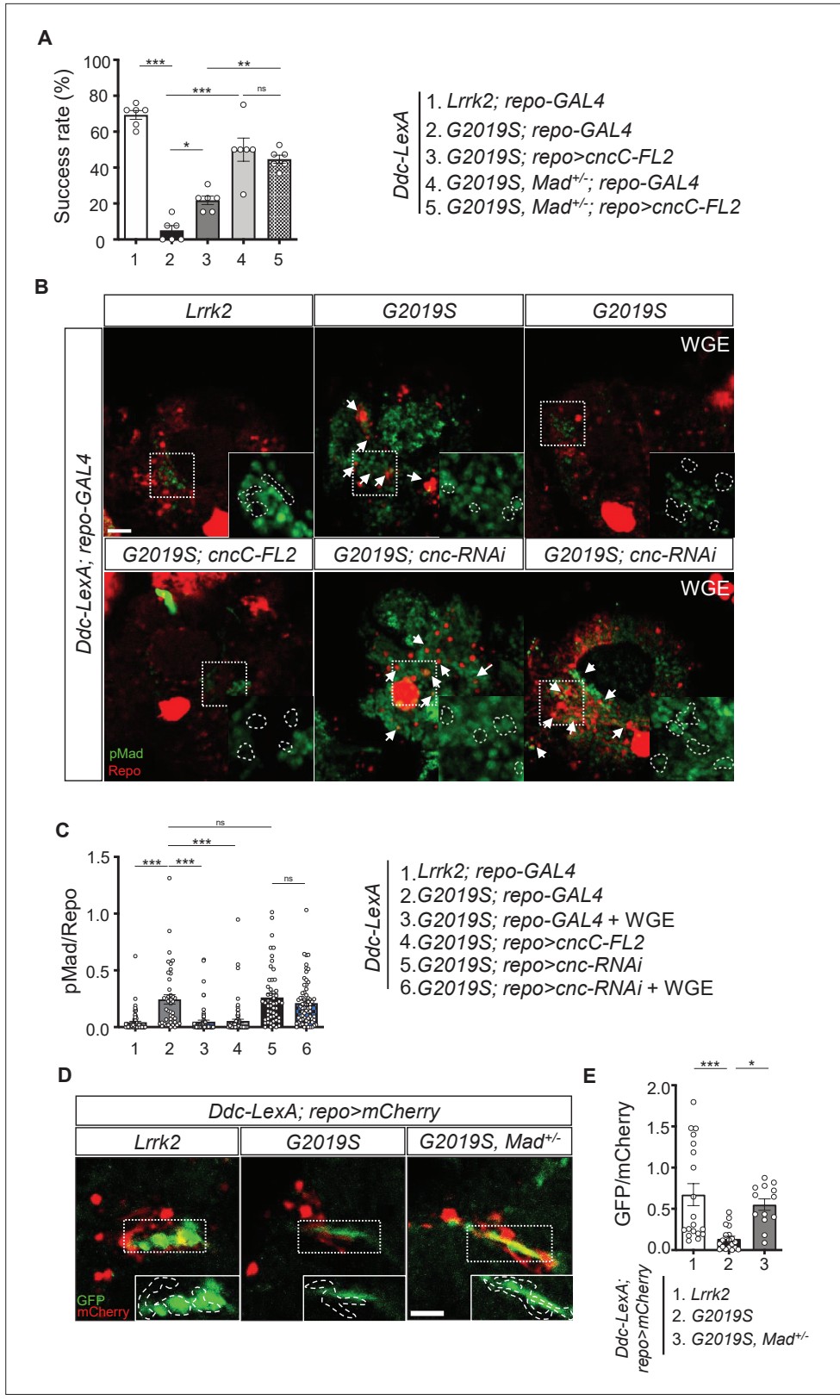

**Figure 8.** Nrf2 antagonizes BMP/Mad signaling in glia. (**A**) Heterozygosity of *Mad* suppresses G2019S mutation-induced locomotion impairment in a negative geotaxis assay. Bar graph shows percentages (mean ± SEM, N = 6) of 6-week-old flies that climbed above 8 cm within 10 s. One-way ANOVA and Tukey's post-hoc multiple comparison test: *p<0.05, **p<0.01, ***p<0.001, ns, not significant. (**B**) Representative images of pMad staining in

*Figure 8 continued on next page*

*Figure 8 continued*

the adult brains of *Ddc-LexA>Lrrk2* or *Ddc-LexA>G2019S* flies in which *repo-GAL4* drives expression of *cncC-FL2* or *cnc-RNAi*. Impact of WGE treatment is shown in the rightmost panels. Arrows indicate pMad and Repo dual-positive cells. Insets are enlarged images of the dashed boxes, and dashed lines encompass Repo-positive cells. Bar: 20 µm. (**C**) Quantification of the ratio of pMad to Repo (mean ± SEM, N > 40). One-way ANOVA and Tukey's post-hoc multiple comparison test: ***p<0.001, ns, not significant. (**D**) Images show pan-glial *ARE-GFP* expression (*repo>mCherry*) in *Ddc-LexA>Lrrk2*, *Ddc-LexA>G2019S*, and *Ddc-LexA>G2019S*, *Mad$^{+/-}$* fly brains. GFP channels within the dashed boxes are shown as enhanced views in the inset, with glial signals outlined by dashed lines. Bar: 20 µm. (**E**) Quantification for GFP expression levels (mean ± SEM, n > 13). One-way ANOVA and Tukey's post-hoc multiple comparison test: *p<0.05, ***p<0.001.

regulating neuronal excitability and participating in phagocytosis and injury-induced inflammation (**Doherty et al., 2009**; **Otto et al., 2018**). Thus, given their proximity to neurons and similar functions, it is not surprising that both types of glia collectively mediate the protective effects of WGE.

Communication between neurons and glia maintains homeostasis, yet also confers the disease state during neurodegeneration. In the *Drosophila* G2019S model, upregulation of the BMP ligand Gbb in dopaminergic neurons activates Mad/Smad signaling in glia, which promotes neuronal degeneration via a feedback mechanism (**Maksoud et al., 2019**). Surprisingly, although the number of dopaminergic neurons in the fly brain is relatively small, the upregulated pMad signal spreads throughout the brain (**Figure 7B**), suggesting that BMP can be disseminated over long distances. In PD patients, higher levels of TGF-β1 have been detected in the striatum and ventricular cerebrospinal fluid (**Vawter et al., 1996**). Thus, members of the TGF-β1 superfamily such as TGF-β1 and BMP signaling molecules may represent indicators of neuronal degeneration. Accordingly, disrupting the glia-to-neuron feedback mechanism may sustain neuronal survival. In glia, we found that WGE treatment downregulated the pMad levels that had been increased in the G2019S flies (**Figure 7A and B**). Nrf2 activation in glia also suppressed the enhanced levels of pMad in G2019S flies (**Figure 8B and C**). Indeed, our genetic assays indicate that the Nrf2 and Mad pathways interact in the glia of G2019S flies. Thus, WGE exerts its beneficial effects by activating Nrf2 to antagonize the Mad activity that would otherwise contribute to the degeneration of dopaminergic neurons. As a transcriptional activator, Nrf2 induces expression of the inhibitory component Smad7 to form inactive Smad complexes (**Song et al., 2019**) and the phosphatase subunit PPM1A to alter Smad2/3/4 phosphorylation and DNA binding (**Lin et al., 2006**). Given that these components are conserved in *Drosophila*, Nrf2 may employ similar pathways to block Mad signaling in glia.

That glial Nrf2 activation protects neurons is evidenced by our observations of enhanced HO-1 expression (**Figure 4C**) and increased numbers of dopaminergic neurons (**Figure 5C and D**). These results support that the role of Nrf2 in glia is to induce expression of antioxidation building blocks, such as phase-II detoxification enzymes, and to enhance inflammatory processes (**Hirrlinger and Dringen, 2010**; **Rojo et al., 2010**). In a model of fibrosis, TGF-β/Smad2/3 suppressed expression of the *ARE*-luciferase reporter and glutathione (**Ryoo et al., 2014**). Moreover, Nrf2 knockdown was shown to reduce expression of the antioxidative enzyme NAD(P)H quinone dehydrogenase 1 (NQO1), thereby elevating cellular oxidative stress and upregulating TGF-β/Smad targeted gene expression (**Prestigiacomo and Suter-Dick, 2018**). Hence, we propose that WGE promotes Nrf2 activation to antagonize the Smad signaling in glia that is induced by dopaminergic neuron-secreted BMP/Gbb signal during degeneration.

In our study, the *LRRK2-G2019S* mice show locomotor defects and dopaminergic loss at the age of 11.5 months. A previous study shows only earlier signs of defects, the reduction of the dopamine level and release at the age of 12 months in the *LRRK2-G2019S* mice (**Li et al., 2010**), which could be contributed by the genetic background (FVB/NJ vs. C57BL/6J). Nevertheless, we have demonstrated that feeding these mice with WGE rescues their locomotor coordination, suppresses their centrophobism, and recovers their numbers of dopaminergic neurons and hyperactivated microglia (**Figure 9** and **Figure 10** A – D). Significantly, we found that the activity of the TGF-β/Smad2/3 pathway was elevated in nigrostriatal brain lysates, and this activity was also suppressed by WGE treatment (**Figure 10**). Collectively, these results from fly and mouse PD models indicate that the effectiveness of WGE is likely mediated through conserved Nrf2/Mad pathways (**Figure 11**). Our findings contribute

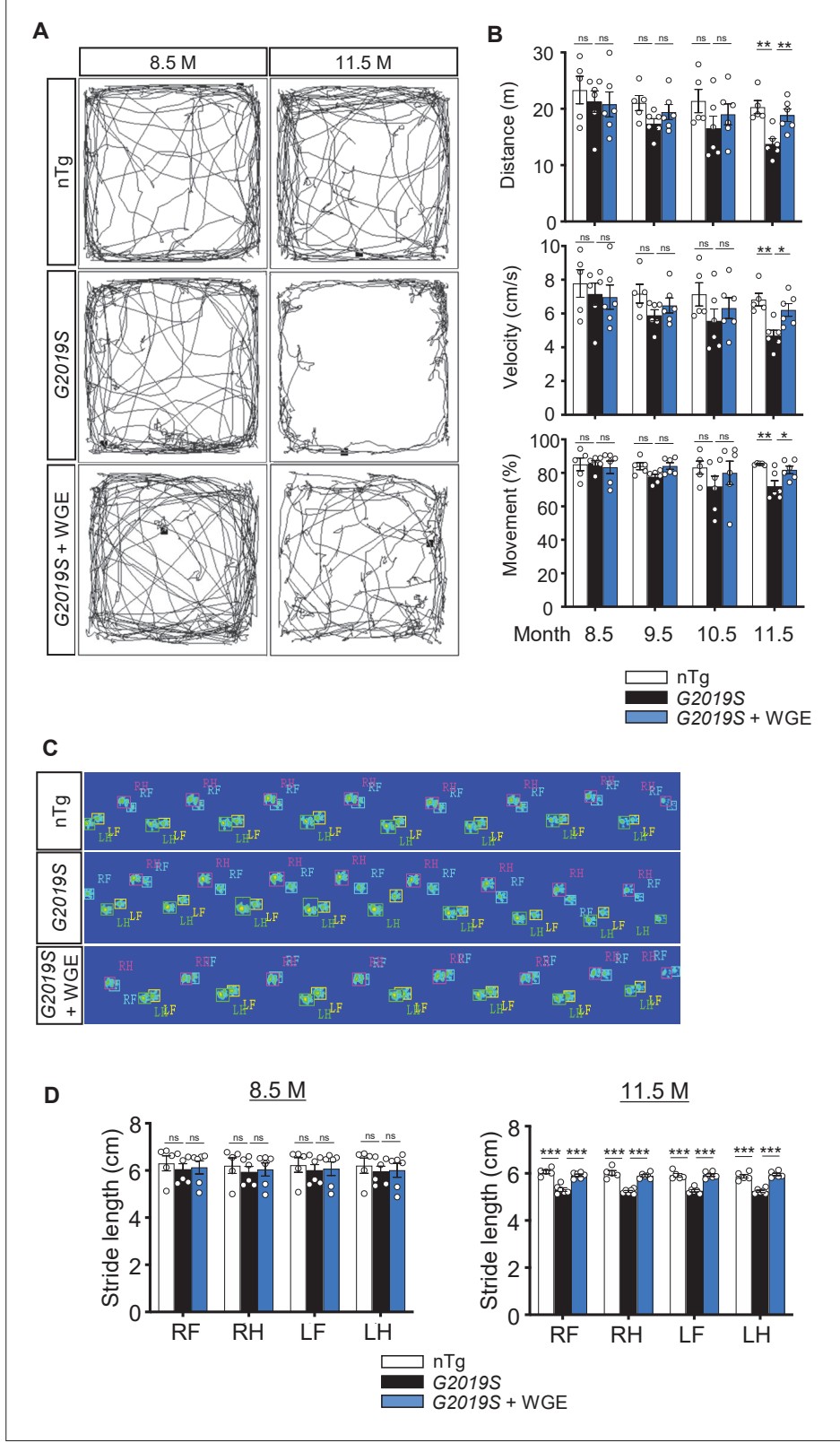

**Figure 9.** Water extract of *Gastrodia elata* Blume (WGE) treatment rescues impaired locomotion of *LRRK2-G2019S* mice. (**A**) Video-tracked paths for nTg, *LRRK2-G2019S*, and WGE-fed *LRRK2-G2019S* mice (8.5 and 11.5 months old) during the open-field test. (**B**) Quantification of total distance (m), average velocity (cm/s), and percentage moving time for 8.5-, 9.5-, 10.5-, and 11.5-month-old mice. (**C**) Captured and converted images of single stance

*Figure 9 continued on next page*

*Figure 9 continued*

for each paw of 11.5-month-old nTg, *LRRK2-G2019S,* and WGE-fed *LRRK2-G2019S* mice in a catwalk analysis. (**D**) Quantification of stride length for each paw of 8.5- and 11.5-month-old mice. Data in (**B**) and (**D**) are presented as mean ± SEM (nTg; N = 5, *G2019S*; N = 6, WGE-fed *G2019S*; N = 6). One-way ANOVA and Tukey's post-hoc multiple comparison test: *p<0.05, **p<0.01, ***p<0.001, ns, not significant. RF, right front; RH, right hind; LF, left front; LH, left hind.

to our mechanistic understanding of PD and provide potential therapeutic strategies that incorporate the traditional herbal medicine GE.

# Materials and methods
## *Drosophila* stocks and maintenance
All fly stocks were maintained on standard cornmeal-based food medium at 25°C. *Drosophila* stocks sourced from the Bloomington *Drosophila* Stock Center (Indiana University, Bloomington, USA) were *UAS-mCD8-GFP* (#5137), *elac-GAL4* (#8760), *repo-GAL4* (#7215), *Ddc-LexA* (#54218), *UAS-cncTRIP* (#25984), *Tub-GAL80^{ts}* (#7108), *alrm-GAL4* (#67032), *R56F03-GAL4* (#39157), *UAS-tkv^{Q253D}* (#36536), and *Mad^{K00237}* (#10474). *NP2222-GAL4* (#112830) was from the *Drosophila* Genomics Resource Center and *NP6293-GAL4* (#105188) was from the Kyoto Stock Center. Other stocks include *UAS-Flag-LRRK2-WT* (*Lin et al., 2010*), *UAS-Flag-LRRK2-G2019S* (*Lin et al., 2010*), *Ddc>GAL4* (*Sang et al., 2007*), *ARE-GFP* (*Sykiotis and Bohmann, 2008*), *UAS-cncC-FL2* (*Sykiotis and Bohmann, 2008*), *moody-GAL4* (*Bainton et al., 2005*), and *UAS-Mad* (*Takaesu et al., 2006*). The two *LexAop* fly lines—*LexAop-LRRK2-WT* and *LexAop-LRRK2-G2019S*—were generated in this study. In brief, the cDNAs for *LRRK2-WT* and *LRRK2-G2019S* were isolated from the pDEST53-LRRK2-WT and pDEST53-LRRK2-G2019S plasmids (Addgene, MA) for subcloning into *LexAop* plasmids (Addgene), which were for microinjection (Fly facility, University of Cambridge, UK). The transgenes were site landed at an attP site on the 2nd chromosome (25C6). For temperature-shift assay of *GAL80^{ts}* flies, parental flies were maintained at 19°C and allowed to mate, before collecting the F1 adults and shifting them to 29°C to inactivate GAL80.

## Preparation of WGE and related chemical compounds
Authentication of GE and preparation of WGE were as described previously (*Lin et al., 2018*; *Lin et al., 2016a*). WGE (KO DA Pharmaceutical Co. Ltd., Taoyuan, Taiwan), gastrodin (Wuhan YC Fine Chemical Co., Wuhan, China), and 4-HBA (Sigma-Aldrich, Darmstadt, Germany) were added to freshly prepared cornmeal-based fly food at indicated final concentrations (w/w). For experiments, 1- to 3-day-old post-eclosion flies were collected and transferred to fresh food medium twice per week.

## Fly locomotion assay
A negative geotaxis climbing assay was performed to assess locomotor activity, and it was conducted according to a previous study with minor modification (*Madabattula et al., 2015*). Cohorts of 35 flies from each genotype were assayed weekly for six consecutive weeks. Success rates were calculated as the percentage of flies that could climb above the 8 cm mark of a 20 cm cylinder within 10 s.

The free-walking assay protocol was conducted based on a previous report with minor modification (*Chen et al., 2014*). Cohorts of eight flies were habituated on a 10 cm agar-filled dish for 30 min. The dishes were gently tapped to encourage the flies to walk, which was video-taped for 5 min. Movement tracks were processed in ImageJ and quantified using the Caltech multiple fly tracker (Ctrax).

## Immunostaining and immunoblotting of adult fly brains
The protocol for immunostaining whole-mount adult brains was essentially as described previously (*Lin et al., 2010*; *Maksoud et al., 2019*). Adult fly brains for each genotype were dissected at the indicated timepoints for immunostaining with the following primary antibodies: mouse anti-TH (Immunostar, 22941, 1:1000), mouse anti-repo (Hybridoma Bank DSHB, 8D12, 1:500), chicken anti-GFP (Abcam, ab13970, 1:10,000), and rabbit anti-phospho-Smad3 (Ser423/425) (Abcam, ab52903, 1:250) (*Smith et al., 2012*). Fluorophore-conjugated secondary antibodies were FITC-conjugated goat anti-mouse IgG (Jackson ImmunoResearch, AB_2338589, 1:500), Alexa Fluor 488-conjugated goat anti-mouse

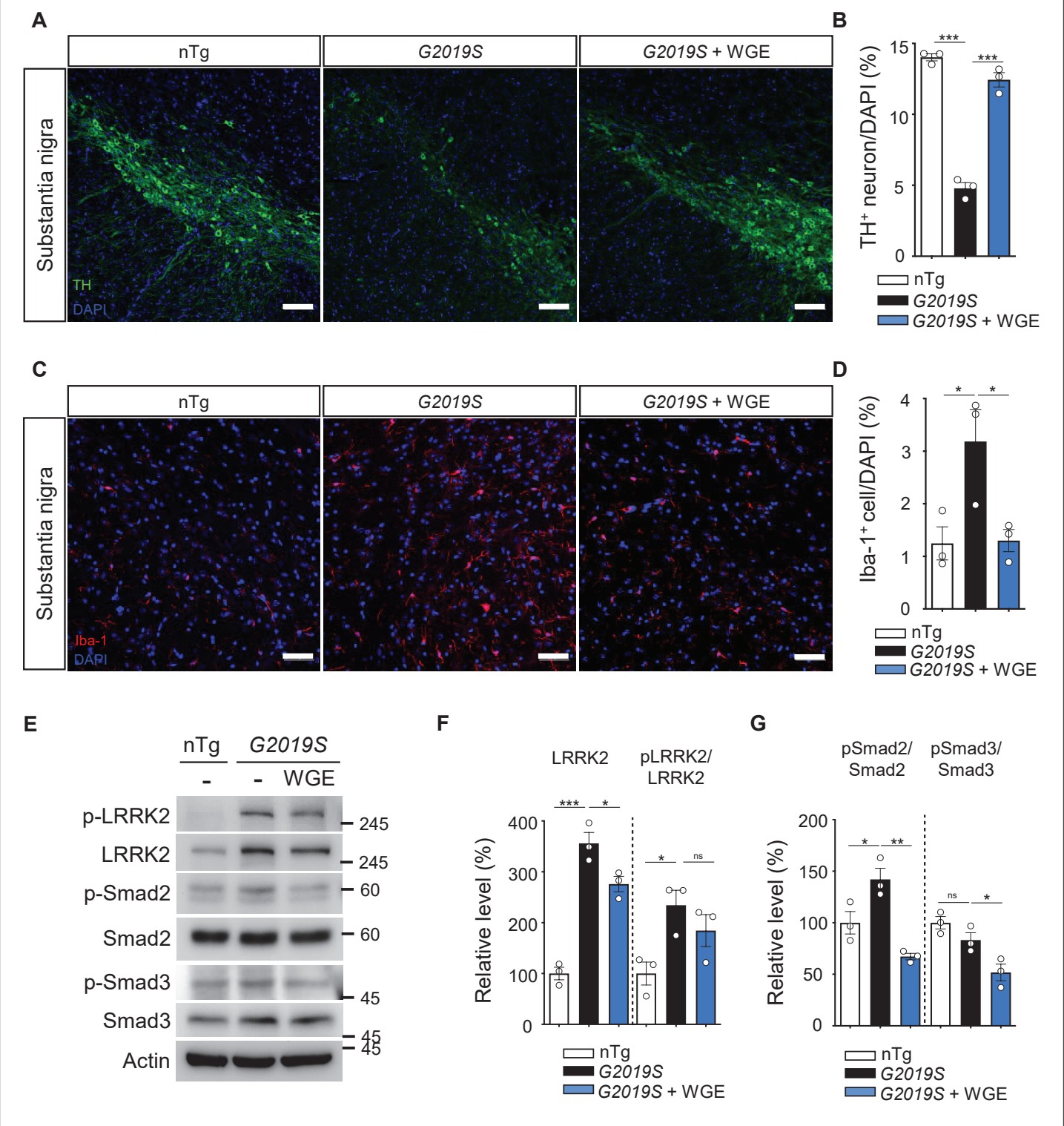

**Figure 10.** Water extract of *Gastrodia elata* Blume (WGE) prevents dopaminergic neuron loss, microglial activation, and phosphorylation of LRRK2, Smad2, and Smad3. (**A, C**) Representative images showing TH-positive dopaminergic neurons (**A**) or Iba-1-positive microglia (**C**) in the substantia nigra of 11.5-month-old nTg, *LRRK2-G2019S,* and WGE-fed *LRRK2-G2019S* mice. Bars: 100 µm in (**A**) and 50 µm in (**C**). (**B, D**) Quantification of numbers of TH-positive (**B**) or Iba-1-positive cells (**D**) relative to DAPI cells. (**E**) Representative immunoblots of nigrostriatal lysates prepared from 11.5-month-old nTg, *LRRK2-G2019S,* and WGE-fed *LRRK2-G2019S* mice reveal expression levels of LRRK2, pLRRK2 (Ser[1292]), Smad2, pSmad2 (Ser[465] and Ser[467]), Smad3, and pSmad3 (Ser[423] and Ser[425]). Actin acted as a loading control. (**F, G**) Quantifications of LRRK2 and pLRRK2/LRRK2 (**F**), as well as pSmad2/Smad2 and pSmad3/Smad3 (**G**). Data in (**B, D, F, G**) are presented as mean ± SEM (N = 3). One-way ANOVA and Tukey's post-hoc multiple comparison test:

*Figure 10 continued on next page*

*Figure 10 continued*

*p<0.05, **p<0.01, ***p<0.001, ns, not significant.

The online version of this article includes the following figure supplement(s) for figure 10:

**Figure supplement 1.** Water extract of *Gastrodia elata* Blume (WGE) suppresses microglia activation in *LRRK2-G2019S* mice.

IgG (Invitrogen, A28175, 1:500), Cy3-conjugated goat anti-mouse IgG (Jackson ImmunoResearch, AB_2338680, 1:500), Alexa Fluor 488-conjugated goat anti-rabbit IgG (Invitrogen, A27034, 1:500), and Cy5-conjugated goat anti-rat IgG (Invitrogen, A10525, 1:500). Phalloidin-TRITC (Sigma-Aldrich, P1951, 1:5000) that binds F-actin was also used for counterstaining. Immunofluorescence signals were acquired under confocal microscopy (ZEISS LSM 710, Germany).

Adult brain extracts were prepared according to a previously described protocol (*Lin et al., 2010*). In brief, ~80 fly heads for each genotype were isolated for extract preparation. Equivalent amounts of samples (30 µg/20 µL/well) were resolved by SDS-PAGE for immunoblotting. The following primary antibodies were used: rabbit anti-human LRRK2 (Abcam, ab133474, 1:1000); rabbit anti-phospho-LRRK2 (Ser$^{1292}$) (Abcam, ab203181, 1:500); rabbit anti-Akt (Cell Signaling, #4691, 1:1000); rabbit anti-*Drosophila* phospho-Akt Ser$^{505}$ (Cell Signaling, #4054, 1:500); rabbit anti-Nrf2 (Thermo Fisher Scientific, 710574, 1:1000); rabbit anti-phospho-Nrf2 (Ser$^{40}$) (Thermo Fisher Scientific, PA5-67520, 1:1000); mouse anti-HO-1-1 (Thermo Fisher Scientific, MA1-112, 1:1000); rabbit anti-GAPDH

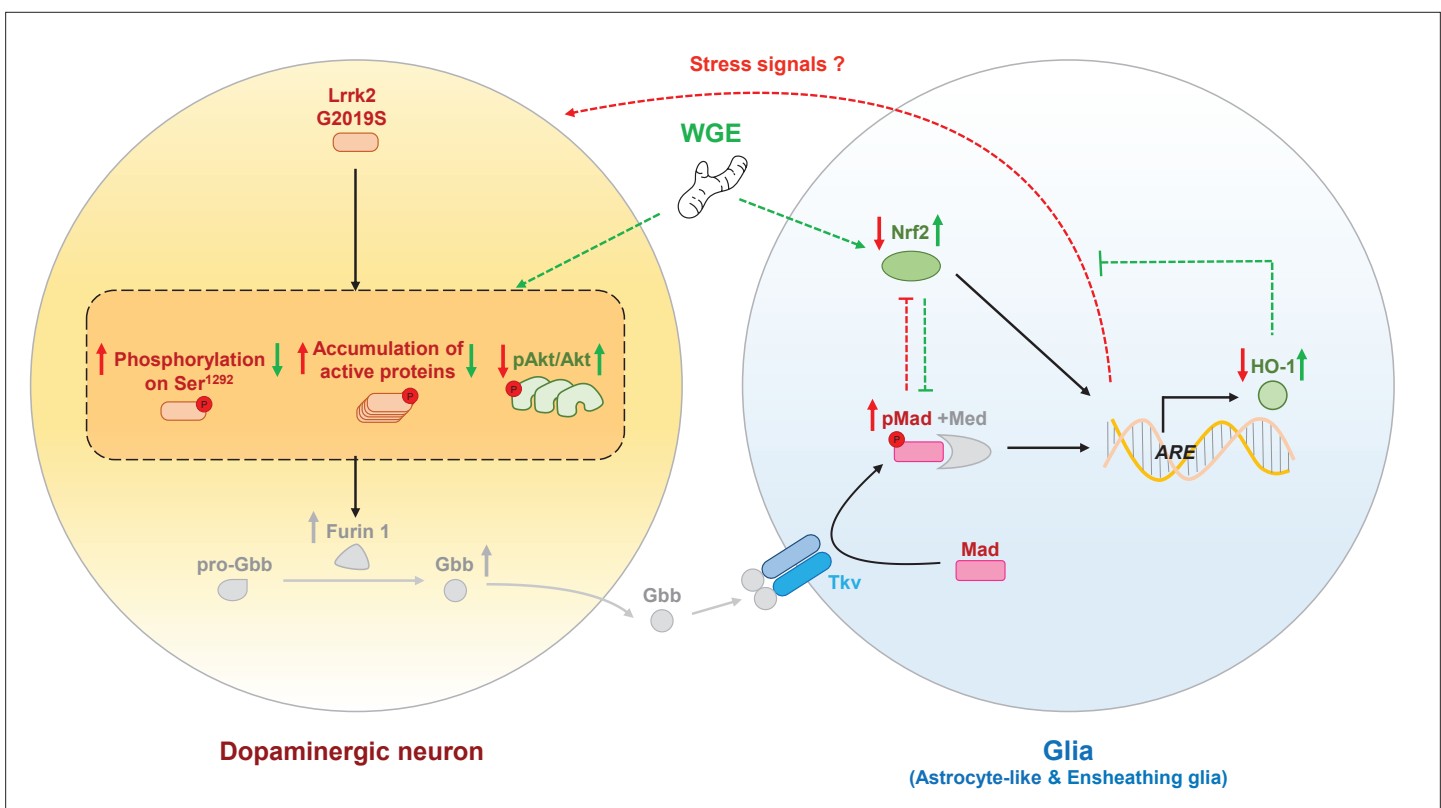

**Figure 11.** The proposed model of water extract of *Gastrodia elata* Blume (WGE) in the G2019S-induced neurodegeneration. Accumulation of the hyperactivated G2019S mutant protein enhances the BMP ligand (Gbb) maturation via upregulation of Furin 1 translation in dopaminergic neurons. Secreted Gbb binds to the BMP receptor, Tkv, and turns on Mad signaling in glia. The G2019S mutation also decreases the Nrf2 activity in the brain, particularly in glia. Both upregulated Mad and downregulated Nrf2 pathways contribute to neurodegeneration. WGE feeding suppresses G2019S hyperactivation in neurons and restores Nrf2 activity mostly in the astrocyte-like and ensheathing glia. WGE-elevated Nrf2 activity in glia antagonizes the BMP/Mad signaling and initiates Nrf2/HO-1 axis in glia, attenuating the stress signals from glia and promoting neuroprotection. Red and green solid arrows (→) indicate the observed effects exerted by G2019S overexpression and WGE feeding, respectively, in the present study. Red and green dashed arrows (-->) indicate the proposed actions trigged by Mad signaling and WGE feeding, respectively. Red and green blunt-ended lines (---|) indicate the proposed inhibitions by Mad and Nrf2 overexpression, respectively. (P) Indicates phosphorylation. The Furin 1-mediated Gbb pathway labeled in gray is modified from **Figure 6** of *Maksoud et al., 2019*.

(GeneTex, GTX100118, 1:5000); and rabbit anti-alpha tubulin (Cell Signaling, #2144, 1:10,000), followed by blotting with secondary antibodies peroxidase-conjugated goat anti-rabbit IgG (Jackson ImmunoResearch, AB_2307391, 1:7500) and peroxidase-conjugated goat anti-mouse IgG (Jackson ImmunoResearch, AB_10015289, 1:7500).

## ARE-GFP reporter assay

The antioxidant response element (ARE)-GFP reporter assay was a modification of the protocol from a previous study (Sykiotis and Bohmann, 2008). ARE-GFP flies of 1 week old (for Figure 5—figure supplement 1) or 6 weeks old (Figure 6) were fed with regular food or food containing 0.1% WGE prior to brain dissection and GFP immunostaining. To quantify ARE-GFP in the 6-week-old adult fly brain, confocal images were processed in ImageJ. Mean GFP fluorescence intensities in mCherry-labeled glial cells and TH-positive dopaminergic neurons of the PPL1 cluster were quantified and normalized as GFP/mCherry.

## Animal care and treatments

Transgenic LRRK2-G2019S mice were purchased from the Jackson Laboratory (JAX stock #009609, Bar Harbor, ME) and maintained at the animal center of the National Taiwan University Hospital (NTUH). Non-transgenic (nTg) and heterozygous transgenic LRRK2-G2019S mice were obtained by crossing heterozygous LRRK2-G2019S mice with wild-type FVB/NJ mice (JAX stock #001800). Mice at 8.5 months old were assigned to one of three groups (5–6 mice per group): nTg, LRRK2-G2019S, and LRRK2-G2019S fed with WGE (0.5 g/kg body weight per day) (Lin et al., 2018) for 3 months.

## Behavioral assays

We employed two behavioral tests to assay mouse motor function, that is, an open-field assay to assess spontaneous locomotor activity and CatWalk XT gait analysis to assay coordination. Behavioral experiments were conducted blind to genotype, as described previously (Lin et al., 2020).

## Immunohistochemical staining

After 3 months of WGE treatment, mice were sacrificed at the age of 11.5 months. The substantia nigra and striatum were dissected out. The substantia nigra was subjected to immunostaining, as described previously (Lin et al., 2020). Anti-tyrosine hydroxylase (TH) (Millipore, AB152, 1:200) and anti-ionized calcium-binding adapter molecule 1 (Iba-1) (GeneTex, GTX100042, 1:200) were used as primary antibodies for 24 hr at 4°C. Secondary antibodies were DyLight 488 goat anti-rabbit 1:300 and Alexa Fluor 546 goat anti-rabbit at 1:200 (25°C for 1 hr). Mounting medium with DAPI (GeneTex, GTX30920) was used as a counterstain.

## Immunoblotting

Frozen nigrostriatal brain tissues were homogenized and mixed with lysis buffer to determine protein content and for immunoblotting, as described previously (Lin et al., 2020). The membrane was incubated overnight at 4°C with the following primary antibodies: anti-LRRK2 (Abcam, ab133474, 1:5000), anti-phospho-LRRK2 (Ser[1292], Abcam, ab203181, 1:1000), anti-Smad2 (Cell Signaling, #5339, 1:1000), anti-phospho-Smad2 (Ser465/467, Cell Signaling, #3108, 1:1000), anti-Smad3 (Cell Signaling, #9523, 1:1000), anti-phospho-Smad3 (Ser423/425, Abcam, ab52903, 1:1000), and anti-beta actin (Sigma-Aldrich, A5441, 1:5000). After washing, peroxidase-conjugated goat anti-rabbit IgG (GeneTex, GTX213110-01, 1:5000) or peroxidase-conjugated goat anti-mouse IgG (GeneTex, GTX213111-01, 1:5000) were used as secondary antibodies.

## Statistical analysis

All statistical analyses were carried out in GraphPad Prism 6 (La Jolla, CA). Data are presented as mean ± standard error (SEM). Statistical analysis was performed using either Student t-test or one-way analysis of variance (ANOVA) followed by Tukey's multiple comparison test. $p < 0.05$ were considered indicative of significance. For exact n numbers, p-values, F-values, t-values, and degrees of freedom of each statistical test, please see the statistical information in Source data 1.

## Acknowledgements

We thank D Bohmann (URMC), U Heberlein (HHMI), SJ Newfeld (ASU), T-K Sang (NTHU), Bloomington Stock Center, *Drosophila* Genomics Resource Center, Kyoto Stock Center, and Taiwan Fly Core for providing fly stocks. We thank KO DA Pharmaceutical Co. Ltd. (Taoyuan, Taiwan) and Wuhan YC Fine Chemical Co. (Wuhan, China) for providing WGE and gastrodin, respectively.

## Additional information

### Funding

| Funder | Grant reference number | Author |
|---|---|---|
| Ministry of Science and Technology, Taiwan | MOST-108-2311-B-001-039-MY3 | Cheng-Ting Chien |
| Parkinson's UK | H1201 | Stavroula Petridi Christopher JH Elliott |

The funders had no role in study design, data collection and interpretation, or the decision to submit the work for publication.

### Author contributions

Yu-En Lin, Conceptualization, Data curation, Formal analysis, Investigation, Methodology, Validation, Visualization, Writing – original draft, Writing – review and editing; Chin-Hsien Lin, Conceptualization, Investigation, Project administration, Resources, Supervision, Validation, Writing – review and editing; En-Peng Ho, Yi-Ci Ke, Data curation, Formal analysis, Methodology; Stavroula Petridi, Christopher JH Elliott, Resources; Lee-Yan Sheen, Conceptualization, Investigation, Resources, Supervision, Validation, Writing – review and editing; Cheng-Ting Chien, Conceptualization, Funding acquisition, Investigation, Project administration, Resources, Supervision, Validation, Writing – review and editing

### Author ORCIDs

Yu-En Lin ⓘ http://orcid.org/0000-0002-5848-5405
Chin-Hsien Lin ⓘ http://orcid.org/0000-0001-8566-7573
Cheng-Ting Chien ⓘ http://orcid.org/0000-0002-7906-7173

### Ethics

All animal procedures were approved by the local ethics committee and the Institutional Animal Care and Use Committee (IACUC) of the National Taiwan University (IACUC approval no. 20180103).

### Decision letter and Author response

Decision letter https://doi.org/10.7554/eLife.73753.sa1
Author response https://doi.org/10.7554/eLife.73753.sa2

## Additional files

### Supplementary files

• Transparent reporting form

• Source data 1. Statistical information.
The file includes all data and the statistical analyses in this article.

• Source data 2. Data—western blotting.
The file includes the uncropped images of the western blotting in this article.

### Data availability

All data generated or analyzed during this study are included in the manuscript and supporting files.

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

# Appendix 1

**Appendix 1—key resources table**

| Reagent type (species) or resource | Designation | Source or reference | Identifiers | Additional information |
|---|---|---|---|---|
| Genetic reagent (*Drosophila melanogaster*) | *UAS-mCD8-GFP* | Bloomington *Drosophila* Stock Center | BDSC Cat# 5137; RRID:BDSC_5137 | |
| Genetic reagent (*D. melanogaster*) | *UAS-cncTRiP* | Bloomington *Drosophila* Stock Center | BDSC Cat# 25984; RRID:BDSC_25984 | |
| Genetic reagent (*D. melanogaster*) | *UAS-tkv$^{Q253D}$* | Bloomington *Drosophila* Stock Center | BDSC Cat# 36536; RRID:BDSC_36536 | |
| Genetic reagent (*D. melanogaster*) | *UAS-Flag-LRRK2-WT* | **Lin et al., 2010** | N/A | |
| Genetic reagent (*D. melanogaster*) | *UAS-Flag-LRRK2-G2019S* | **Lin et al., 2010** | N/A | |
| Genetic reagent (*D. melanogaster*) | *UAS-cncC-FL2* | **Sykiotis and Bohmann, 2008** | N/A | |
| Genetic reagent (*D. melanogaster*) | *UAS-Mad* | **Takaesu et al., 2006** | N/A | |
| Genetic reagent (*D. melanogaster*) | *elav-GAL4* | Bloomington *Drosophila* Stock Center | BDSC Cat# 8760; RRID:BDSC_8760 | |
| Genetic reagent (*D. melanogaster*) | *repo-GAL4* | Bloomington *Drosophila* Stock Center | BDSC Cat# 7215; RRID:BDSC_7215 | |
| Genetic reagent (*D. melanogaster*) | *alrm-GAL4* | Bloomington *Drosophila* Stock Center | BDSC Cat# 67032; RRID:BDSC_67032 | |
| Genetic reagent (*D. melanogaster*) | *R56F03-GAL4* | Bloomington *Drosophila* Stock Center | BDSC Cat# 39157; RRID:BDSC_39157 | |
| Genetic reagent (*D. melanogaster*) | *NP2222-GAL4* | Bloomington *Drosophila* Stock Center | BDSC Cat# 112830; RRID:BDSC_112830 | |
| Genetic reagent (*D. melanogaster*) | *NP6293-GAL4* | Kyoto Stock Center | DGRC Cat# 105188; RRID:Kyoto Stock Center_105188 | |
| Genetic reagent (*D. melanogaster*) | *Ddc-GAL4* | **Sang et al., 2007** | N/A | |
| Genetic reagent (*D. melanogaster*) | *moody-GAL4* | **Bainton et al., 2005** | N/A | |
| Genetic reagent (*D. melanogaster*) | *Tub-GAL80$^{ts}$* | Bloomington *Drosophila* Stock Center | BDSC Cat# 7108; RRID:BDSC_7108 | |
| Genetic reagent (*D. melanogaster*) | *Ddc-LexA* | Bloomington *Drosophila* Stock Center | BDSC Cat# 54218; RRID:BDSC_54218 | |
| Genetic reagent (*D. melanogaster*) | *Mad$^{K00237}$* | Bloomington *Drosophila* Stock Center | BDSC Cat# 10474; RRID:BDSC_10474 | |

*Appendix 1 Continued on next page*

*Appendix 1 Continued*

| Reagent type (species) or resource | Designation | Source or reference | Identifiers | Additional information |
|---|---|---|---|---|
| Genetic reagent (*D. melanogaster*) | *ARE-GFP* | ***Sykiotis and Bohmann, 2008*** | N/A | |
| Genetic reagent (*D. melanogaster*) | *LexAop-LRRK2-WT* | This paper | N/A | See Materials and methods, '*Drosophila* stocks and maintenance' |
| Genetic reagent (*D. melanogaster*) | *LexAop-LRRK2-G2019S* | This paper | N/A | See Materials and methods, '*Drosophila* stocks and maintenance' |
| Genetic reagent (*Mus musculus*) | FVB/NJ | The Jackson Laboratory | JAX stock #001800 | |
| Genetic reagent (*M. musculus*) | FVB/N-Tg (LRRK2* G2019S)1Cjli/J | The Jackson Laboratory | JAX stock #009609 | |
| Antibody | Anti-TH (mouse monoclonal) | Immunostar | Cat# 22941; RRID:AB_572268 | IF (1:1000) |
| Antibody | Anti-TH (rabbit polyclonal) | Millipore | Cat# AB152; RRID:AB_390204 | Mouse-IHC (1:200) |
| Antibody | Anti-Repo (mouse monoclonal) | Hybridoma Bank DSHB | Cat# 8D12; RRID:AB_528448 | IF (1:500) |
| Antibody | Anti-GFP (chicken polyclonal) | Abcam | Cat# ab13970; RRID:AB_300798 | IF (1:10000) |
| Antibody | Anti-LRRK2 (rabbit monoclonal) | Abcam | Cat# ab133474; RRID:AB_2713963 | Fly-WB (1:1000) mouse-WB (1:5000) |
| Antibody | Anti-LRRK2 (phospho Ser1292) (rabbit monoclonal) | Abcam | Cat# ab203181 | Fly-WB (1:500) mouse-WB (1:1000) |
| Antibody | Anti-Akt (rabbit monoclonal) | Cell Signaling | Cat# 4691; RRID:AB_915783 | WB (1:1000) |
| Antibody | Anti-phospho-*Drosophila* Akt (Ser505) (rabbit polyclonal) | Cell Signaling | Cat# 4054; RRID:AB_331414 | WB (1:500) |
| Antibody | Anti-Nrf2 (rabbit polyclonal) | Thermo Fisher Scientific | Cat# 710574; RRID:AB_2532742 | WB (1:1000) |
| Antibody | Anti-phospho-Nrf2 (Ser40) (rabbit polyclonal) | Thermo Fisher Scientific | Cat# PA5-67520; RRID:AB_2691678 | WB (1:1000) |
| Antibody | Anti-HO-1-1 (mouse monoclonal) | Thermo Fisher Scientific | Cat# MA1-112; RRID:AB_2536823 | WB (1:1000) |
| Antibody | Anti-GAPDH (rabbit polyclonal) | GeneTex | Cat# GTX100118; RRID:AB_1080976 | WB (1:5000) |
| Antibody | Anti-alpha tubulin (rabbit polyclonal) | Cell Signaling | Cat# 2144; RRID:AB_2210548 | WB (1:10,000) |
| Antibody | Anti-Smad2 (rabbit monoclonal) | Cell Signaling | Cat# 5339; RRID:AB_10626777 | Mouse-WB (1:1000) |

*Appendix 1 Continued on next page*

*Appendix 1 Continued*

| Reagent type (species) or resource | Designation | Source or reference | Identifiers | Additional information |
|---|---|---|---|---|
| Antibody | Anti-phospho-Smad2 (Ser465/467) (rabbit monoclonal) | Cell Signaling | Cat# 3108; RRID:AB_490941 | Mouse-WB (1:1000) |
| Antibody | Anti-Smad3 (rabbit monoclonal) | Cell Signaling | Cat# 9523; RRID:AB_2193182 | Mouse-WB (1:1000) |
| Antibody | Anti-Smad3 (phospho S423 + S425) (rabbit monoclonal) | Abcam | Cat# ab52903; RRID:AB_882596 | IF (1:250) mouse-WB (1:1000) |
| Antibody | Anti-beta actin (mouse monoclonal) | Sigma-Aldrich | Cat# A5441; RRID:AB_476744 | Mouse-WB (1:5000) |
| Antibody | Anti-Iba-1 (rabbit polyclonal) | GeneTex | Cat# GTX100042; RRID:AB_1240434 | Mouse-IHC (1:200) |
| Antibody | Anti-mouse Alexa 488 (goat polyclonal) | Invitrogen | Cat# A28175; RRID:AB_2536161 | IF (1:500) |
| Antibody | Anti-rabbit Alexa 488 (goat polyclonal) | Invitrogen | Cat# A27034; RRID:AB_2536097 | IF (1:500) |
| Antibody | Anti-rabbit DyLight 488 (goat polyclonal) | Thermo Fisher Scientific | Cat# 35552; RRID:AB_844398 | Mouse-IHC (1:300) |
| Antibody | Anti-rabbit Alexa 546 (goat polyclonal) | Thermo Fisher Scientific | Cat# A-11035; RRID:AB_2534093 | Mouse-IHC (1:200) |
| Antibody | Anti-mouse FITC (goat polyclonal) | Jackson ImmunoResearch | Cat#115-095-003; RRID:AB_2338589 | IF (1:500) |
| Antibody | Anti-mouse Cy3 (goat polyclonal) | Jackson ImmunoResearch | Cat# 115-165-003; RRID:AB_2338680 | IF (1:500) |
| Antibody | Anti-rat IgG Cy5 (goat polyclonal) | Invitrogen | Cat# A10525; RRID:AB_2534034 | IF (1:500) |
| Antibody | Anti-rabbit peroxidase (goat polyclonal) | Jackson ImmunoResearch | Cat# 111-035-144; RRID:AB_2307391 | WB (1:7500) |
| Antibody | Anti-mouse peroxidase (goat polyclonal) | Jackson ImmunoResearch | Cat# 115-035-003; RRID:AB_10015289 | WB (1:7500) |
| Antibody | Anti-mouse peroxidase (goat polyclonal) | GeneTex | Cat# GTX213111-01; RRID:AB_10618076 | Mouse-WB (1:5000) |
| Antibody | Anti-rabbit peroxidase (goat polyclonal) | GeneTex | Cat# GTX213110-01; RRID:AB_10618573 | Mouse-WB (1:5000) |
| Chemical compound, drug | Phalloidin-TRITC | Sigma-Aldrich | Cat# P1951; RRID:AB_2315148 | IF (1:5000) |
| Chemical compound, drug | WGE | *Lin et al., 2016a*; *Lin et al., 2018* | N/A | KO DA Pharmaceutical Co. Ltd. |
| Chemical compound, drug | Gastrodin | *Lin et al., 2016b* | N/A | Wuhan YC Fine Chemical Co. |
| Chemical compound, drug | 4-HBA | Sigma-Aldrich | Cat# H20806-10G | |
| Recombinant DNA reagent (*Homo sapiens*) | pDEST53-LRRK2-WT (plasmid) | Addgene | Addgene plasmid # 25044; RRID:Addgene_25044 | |

*Appendix 1 Continued on next page*

*Appendix 1 Continued*

| Reagent type (species) or resource | Designation | Source or reference | Identifiers | Additional information |
|---|---|---|---|---|
| Recombinant DNA reagent (*H. sapiens*) | pDEST53-LRRK2-G2019S (plasmid) | Addgene | Addgene plasmid # 25045; RRID:Addgene_25045 | |
| Recombinant DNA reagent (*D. melanogaster*) | pJFRC19-13XLexAop2-IVS- myr::GFP (plasmid) | Addgene | Addgene plasmid # 26224; RRID:Addgene_26224 | |
| Software, algorithm | ImageJ | PMID:22930834 | https://imagej.nih.gov/ij; RRID:SCR_003070 | |
| Software, algorithm | Ctrax | PMID:19412169 | N/A | |
| Software, algorithm | CatWalk XT | *Lin et al., 2020* | N/A | Noldus Information Technology |
| Software, algorithm | Prism 6 | GraphPad | RRID:SCR_002798 | |
| Other | DAPI | GeneTex | Cat# GTX30920 | |

