## [Editor Report]

Your study elucidates the molecular mechanisms by which a Chinese traditional medicine compound, GE, confers neuroprotection in mouse and fly models of LRRK2 G2019S-induced Parkinson's disease and provides an avenue for a potential therapeutic treatment for patients.

---

## [Decision Letter]

**Decision letter after peer review:**

Thank you for submitting your article "Glial Nrf2 signaling mediates the neuroprotection exerted by *Gastrodia elata* Blume in Lrrk2-G2019S Parkinson's disease" for consideration by *eLife*. Your article has been reviewed by 2 peer reviewers, and the evaluation has been overseen by a Reviewing Editor and Suzanne Pfeffer as the Senior Editor. The following individuals involved in review of your submission have agreed to reveal their identity: Fengwei Yu (Reviewer #2).

Essential revisions:

Please address the issues raised by the reviewers with textual changes.

Congratulations on a nice piece of compelling scientific work!

*Reviewer #2 (Recommendations for the authors):*

Can WGE treatment of 3-4-week-old G2019S flies also alleviate locomotive defects despite the onset of PD progression? If the authors have the data available, it would be nice to include in the revised version. However, this experiment is not required for the publication of the manuscript.

In a previous study by the same lab, they show that Nrf2 pathway is upregulated in neurons upon Lovastatin treatment, suggesting a neuronal function of Nrf2. Nrf2 pathway was also recently reported to regulate developmental neurodegeneration (i.e. dendrite pruning) in a cell-autonomous manner. It could be interesting to examine if Lovastatin treatment upregulates Nrf2 activity in neurons or glia in PD flies. The authors may discuss this part or investigate in their future study.

---

## [Author Response]

Reviewer #2 (Recommendations for the authors):Can WGE treatment of 3-4-week-old G2019S flies also alleviate locomotive defects despite the onset of PD progression? If the authors have the data available, it would be nice to include in the revised version. However, this experiment is not required for the publication of the manuscript.

We thank the reviewer for the comment. We had investigated the therapeutic effect of WGE in the PD fly model. In the experiment, WGE feeding of *Ddc>G2019S* flies was started on the first day of the fourth week and climbing activity was assessed at the end of the fourth week. Interestingly, 0.1% WGE is also effective to significantly improve locomotion of *Ddc>G2019S* flies. The protective effect was reduced at week 5 and diminished at week 6 with continuation of WGE feeding. The result was included in the revised manuscript as Figure 1—figure supplement 2. We also revised the descriptions in Results (Lines: 134-140) and Figure legends of Figure 1—figure supplement 2 (Lines: 944-950) as below.

Lines: 134-140

“We also tested the WGE effect on the *Ddc>G2019S* flies that were fed with regular food without WGE for 3 weeks. At week 4, these *Ddc>G2019S* flies also showed a significant improvement in their climbing ability, compared to the age-matched *Ddc>G2019S* flies fed continuously on regular food (Figure 1—figure supplement 2). The effect of improving climbing activity in the WGE-fed *Ddc>G2019S* flies was reduced at week 5 and diminished at week 6, suggesting that WGE feeding starting at earlier stages is important for long-term locomotion improvement.”

Lines: 944-950

“Figure 1—figure supplement 2. Locomotion improvement of *Ddc>G2019S* flies starting WGE feeding at week 4.

The climbing activities of *Ddc>Lrrk2*, *Ddc>G2019S*, and *Ddc>G2019S* with 0.1% WGE feeding at week 4 were assessed at weeks 3, 4, 5 and 6. Bar graphs show success rates (mean ± SEM, N = 6) of flies climbing over 8 cm height in 10 sec. One-way ANOVA and Tukey’s post-hoc multiple comparison test were performed and statistical significance was shown as *** for *p* < 0.001, * for *p* < 0.05 and ns for no significance.”

In a previous study by the same lab, they show that Nrf2 pathway is upregulated in neurons upon Lovastatin treatment, suggesting a neuronal function of Nrf2. Nrf2 pathway was also recently reported to regulate developmental neurodegeneration (i.e. dendrite pruning) in a cell-autonomous manner. It could be interesting to examine if Lovastatin treatment upregulates Nrf2 activity in neurons or glia in PD flies. The authors may discuss this part or investigate in their future study.

We thank the reviewer for the comment. We added a paragraph to address the neuronal function of Nrf2 and lovastatin-related future works in Discussion (Lines: 437-441), as below.

Lines: 437-441

“…In a previous study, lovastatin treatment provides neuroprotection in the G2019S-induced PD model, also through the Akt/Nrf2 pathway (C. H. Lin et al., 2016). As activation of neuronal Nrf2 plays a non-conventional role in promoting developmental dendrite pruning (Chew et al., 2021), it remains interesting to further study the cell types that mediate the action of lovastatin.

Newly included references:

Chew, L. Y., Zhang, H., He, J., and Yu, F. (2021). The Nrf2-Keap1 pathway is activated by steroid hormone signaling to govern neuronal remodeling. *Cell rep*, *36*(5), 109466. https://doi.org/10.1016/j.celrep.2021.109466